# Distributed Nonparametric Estimation: from Sparse to Dense Samples per Terminal

**Deheng Yuan** [1] **Tao Guo** [2] **Zhongyi Huang** [1]

## Abstract

Consider the communication-constrained problem of nonparametric function estimation, in which each distributed terminal holds multiple i.i.d. samples. Under certain regularity assumptions, we characterize the minimax optimal rates for all regimes, and identify phase transitions of the optimal rates as the samples per terminal vary from sparse to dense. This fully solves the problem left open by previous works, whose scopes are limited to regimes with either dense samples or a single sample per terminal. To achieve the optimal rates, we design a layered estimation protocol by exploiting protocols for the parametric density estimation problem. We show the optimality of the protocol using information-theoretic methods and strong data processing inequalities, and incorporating the classic balls and bins model. The optimal rates are immediate for various special cases such as density estimation, Gaussian, binary, Poisson and heteroskedastic regression models.

## 1. Introduction

Distributed nonparametric estimation problems have attracted wide attention, and related theoretical studies can shed light on the understanding of modern applications such as federated learning (McMahan et al., 2017; Li et al., 2020; P. Kairouz, et al., 2021). In this setting, multiple distributed terminals cooperate to estimate a nonparametric function, while each of them can only observe part of samples and use limited number of bits to describe the observation. The limitation of communication resources often leads to an increase in estimation error compared with the classic centralized settings where all the samples are accessed directly.

In this work, we investigate the nonparametric function estimation problem, where each of the $m$ terminals observes $n$ i.i.d. samples and has a communication budget of $l$ bits. We consider all parameter regimes from sparse to dense samples per terminal, i.e., there is no restriction on the relative value of $n$ compared to $m$. Inspired by (Zaman & Szabó, 2022), a unified nonparametric estimation model describing many ways of sample generation is adopted in this work, so that many specific settings are subsumed.

### 1.1. Our Contributions

The main contribution of this work is that we obtain the minimax optimal rates (up to logarithmic factors) for the aforementioned estimation problem, under a few regularity assumptions. The assumptions are satisfied in several key estimation settings, including density estimation, Gaussian, binary, Poisson and heteroskedastic regression models. Hence, the optimal rates for these models follow directly as corollaries of the main results.

Previous works focused on either the case $m < n^\gamma, \gamma < 2r$ ($r$ is the Sobolev regularity parameter) with dense samples (Szabó & van Zanten, 2020; Zaman & Szabó, 2022) or the specific density estimation problem with extremely sparse $n = 1$ sample (Barnes et al., 2020; Acharya et al., 2024). We fully solve the problem by characterizing the optimal rates for all regimes from sparse to dense samples per terminal classified by the relative size of $n$ and $m$. As a result, We see that the dependence of the optimal rates on the communication budget $l$ can be qualitatively different for sparse and dense settings, where phase transitions are clearly characterized.

To establish our results, we need to prove both the upper and lower bounds for the minimax rate. For the upper bound, we design a two-layer estimation protocol. The outer layer transforms the original problem into a parametric density estimation problem, and the inner layer solves it by exploiting the protocol developed in (Yuan et al., 2024). The design of the outer layer employs wavelet-based estimator with sparsity properties, incorporating appropriate

[1]Department of Mathematical Sciences, Tsinghua University, Beijing, China. [2]School of Cyber Science and Engineering, Southeast University, Nanjing, China.. Correspondence to: Tao Guo <taoguo@seu.edu.cn>, Zhongyi Huang <zhongyih@tsinghua.edu.cn>.

*Proceedings of the 42$^{nd}$ International Conference on Machine Learning*, Vancouver, Canada. PMLR 267, 2025. Copyright 2025 by the author(s).

truncation and quantization. Moreover, parameters linking two layers are tuned carefully to achieve optimality. For the lower bound, we take advantage of information-theoretic methods (Barnes et al., 2020; Acharya et al., 2022; 2023). We prove and apply a generalized tensorization of the strong data processing inequality, in which the key step is to bound the strong data processing constant. To establish this bound, we interpret the likelihood ratio by drawing connections to the classic balls and bins model.

## 1.2. Comparisons with Related Works

As a theoretical framework for federated learning, distributed nonparametric estimation problems under communication constraints received wide attention (Zhu & Lafferty, 2018; Barnes et al., 2020; Szabó & van Zanten, 2020; 2022; Cai & Wei, 2022; Zaman & Szabó, 2022; Acharya et al., 2024), as well as related problems under differential privacy constraints (Butucea et al., 2020; Kroll, 2021; Sart, 2023; Lalanne et al., 2023; Cai et al., 2024). For the problems under communication constraints, optimal rates for certain special cases are obtained such as the nonparametric density estimation (Barnes et al., 2020; Acharya et al., 2024), nonparametric Gaussian regression (Szabó & van Zanten, 2020), Gaussian sequence model with white noise (Zhu & Lafferty, 2018; Szabó & van Zanten, 2022; Cai & Wei, 2022) and two-party joint distribution estimation (Liu, 2022; 2023). In (Zaman & Szabó, 2022), a general framework was developed and optimal rates were derived for several special cases under specific assumptions.

In the setting where each terminal has $n$ i.i.d. samples, previous works focused on either the regime $m < n^\gamma, \gamma < 2r$ ($r$ is the Sobolev regularity parameter) with dense samples (Szabó & van Zanten, 2020; Zaman & Szabó, 2022) or the density estimation problem with extremely sparse $n = 1$ sample (Barnes et al., 2020; Acharya et al., 2024). However, the problem for other choices of $n$ is more difficult, since the methods in (Barnes et al., 2020; Acharya et al., 2024; Szabó & van Zanten, 2020; Zaman & Szabó, 2022) cannot be applied without substantial development.

In the current work we set no restrictions on the relative size of $n$ and $m$ and obtain the optimal rates for all regimes, which fully solves the problem. To handle this general case, our framework imposes assumptions which are strengthened slightly from that in (Zaman & Szabó, 2022). All the special cases therein are still subsumed in our setting. Moreover, different from (Zaman & Szabó, 2022), our assumptions are imposed on single random variables, making them easier to verify.

Specialized to regression problems like the nonparametric Gaussian regression, our problem leads to a setting with random design where the explanatory variable is randomly generated. The resulting problem is substantially different

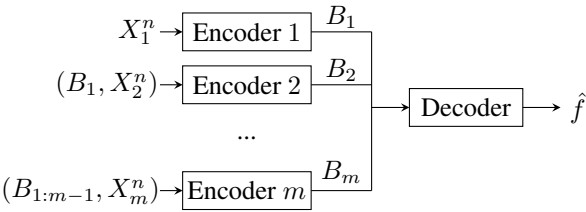

*Figure 1.* Distributed interactive nonparametric estimation

and more difficult than the Gaussian sequence model studied by (Zhu & Lafferty, 2018; Szabó & van Zanten, 2022; Cai & Wei, 2022), and merits separate investigation. The difference is reflected in the optimal rates, which depend exponentially in $l$ for some regimes of our problem, but always polynomially for the Gaussian sequence model.

Moreover, it is interesting to see the similarities in the phase transitions of the optimal rates between the nonparametric estimation problem considered in this work and the parametric distribution estimation problem (Acharya et al., 2020; 2021; Yuan et al., 2024). The inherent connections are partially revealed in this work.

## 1.3. Problem Formulation

We denote a discrete random variable by a capital letter and use the superscript $n$ to denote an $n$-sequence, e.g., $X^n = (X_i)_{i=1}^n$. For any positive functions $a(m, n, l, r)$ and $b(m, n, l, r)$, we say $a \preceq b$ if $a \leq c \cdot b$ for some constant $c > 0$ independent of parameters $(m, n, l)$. The notation $\succeq$ is defined similarly. Then we denote by $a \asymp b$ if both $a \preceq b$ and $a \succeq b$ hold.

Let $p_f$ be an unknown distribution (assume $p_f$ is the pmf or pdf) parameterized by a function $f$ belonging to a subset $\mathcal{F}$ of the standard Sobolev ball $H^r([0, 1], L)$, $r > \frac{1}{2}$, $L > 0$. See Section 3 for a brief introduction of Sobolev spaces and wavelets. Assume that samples are generated at random according to $p_f$. To be precise, let $X_{ij} \sim p_f(x), i = 1, 2, \cdots, m, j = 1, 2, \cdots, n$ be i.i.d. random variables distributed over $\mathcal{X}$. Denote the total sample size by $N = mn$.

Consider the distributed nonparametric minimax estimation problem with communication constraints depicted in Figure 1. Assume there are $m$ encoders. For $i = 1, ..., m$, the $i$-th encoder observes the source message $X_i^n = (X_{ij})_{j=1}^n$ and the first $i-1$ coded messages $(B_{i'})_{i'=1}^{i-1}$. Then the coded binary message $B_i$ of length $l$ is transmitted to the decoder and the remaining $m - i$ encoders. Upon receiving messages $B^m = (B_i)_{i=1}^m$, the decoder needs to establish a reconstruction $\hat{f} \in H^r([0, 1], L)$ of $f$. We assume that $l \geq 4$ in this work, which is reasonable for most cases.

An $(m, n, l)$ sequentially interactive protocol $\mathcal{P}$ is defined

by a series of random encoding functions

$$\text{Enc}_i : \mathcal{X}^n \times \{0,1\}^{(i-1)l} \to \{0,1\}^l, \forall i = 1, ..., m,$$

and a random decoding function

$$\text{Dec} : \{0,1\}^{ml} \to H^r([0,1], L).$$

Then we have $B_i = \text{Enc}_i(X^n, B_{1:i-1})$ and the reconstruction of the function is $\hat{f}_{\mathcal{P}} = \text{Dec}(B_1, B_2, ..., B_m)$.

Define the minimax convergence rate as

$$R(m, n, l, r) = \inf_{(m,n,l)\text{-protocol } \mathcal{P}} \sup_{f \in \mathcal{F}} \mathbb{E}[\|\hat{f}_{\mathcal{P}} - f\|_2^2]. \quad (1)$$

The parameter $L$ is omitted in (1), since we assume that $L$ is a constant and are only interested in the order of the convergence rate $R$ in this work.

## 2. Main Results

For the case $l = \infty$, i.e. there are no communication constraints, it is well known that (cf. Section 6.3.3 in (Giné & Nickl, 2015) and Section 15.3 in (Wainwright, 2019))

$$R(m, n, \infty, r) \asymp N^{-\frac{2r}{2r+1}}$$

for specific settings like density estimation and Gaussian regression. Recall that $N = mn$ is the total sample size.

To see the effect of communication constraints, first define the effective sample size $N_{ess}$ as

$$\left\{ \left[ (2^l mn)^{\frac{2r+1}{2r+2}} \wedge lm \right] \vee \left[ (lm)^{2r+1} \wedge (lmn)^{\frac{2r+1}{2r+2}} \right] \right\} \wedge mn. \quad (2)$$

It is easy to see that for $l = \infty$, we have $N_{ess} = mn$. Further denote by $\text{Poly}(\log N)$ a polynomial of $\log N$. In the following, we show that the optimal rate for the problem in Section 1.3 with total sample size $mn$ is roughly the same as that for the problem without communication constraints with total sample size $N_{ess}$, which is the main theorem of this work.

**Theorem 2.1.** *Under Assumptions 4.1, 5.1 and 5.2, we have* $R(m, n, l, r) \succeq (N_{ess})^{-\frac{2r}{2r+1}}/\text{Poly}(\log N)$ *and* $R(m, n, l, r) \preceq (N_{ess})^{-\frac{2r}{2r+1}}\text{Poly}(\log N)$.

Theorem 2.1 shows that the minimax optimal rate $R(m, n, l, r)$ is approximately $(N_{ess})^{-\frac{2r}{2r+1}}$ under a few necessary assumptions 4.1, 5.1 and 5.2 on the sample distribution $p_f$. Details of assumptions are omitted here and will be formulated in Sections 4 and 5. Assumption 4.1 is for the upper bound, which requires the existence of good estimators for $f$ based on each sample $X \sim p_f$. The remaining assumptions 5.1 and 5.2 are for the lower bound, which reveals the difficulty of estimating $f$ based on samples generated from $p_f$. These assumptions are reasonable, and can be verified for many common examples such as those presented in Section 2.3.

*Proof of Theorem 2.1.* We collect different cases in (2) based on the exact formula of $N_{ess}$. Logarithmic factors in some boundaries that do not affect the conclusion in Theorem 2.1 are omitted. The proof of both lower and the upper bounds needs to handle all the following cases respectively.

1. $N_{ess} = (2^l mn)^{\frac{2r+1}{2r+2}}$ for $m \geq n^{2r+1}$ and $1 \leq l \leq \frac{1}{2r+1}\log\frac{m}{n^{2r+1}}$. In this case $n^{2r+1} \leq N_{ess} \leq m$.

2. $N_{ess} = lm$ for $m > n^{2r}$ and $\frac{1}{2r+1}\log\frac{m}{n^{2r+1}} \vee \frac{n^{2r+1}}{m} \leq l \leq n$. In this case $N_{ess} \geq n^{2r+1} \vee m$.

3. $N_{ess} = (lm)^{2r+1}$ for $n > m^{2r+1}$ and $1 \leq l \leq \frac{n^{\frac{1}{2r+1}}}{m}$. In this case $N_{ess} \leq n$.

4. $N_{ess} = (lmn)^{\frac{2r+1}{2r+2}}$ for $m < n^{2r+1}$ and $\frac{n^{\frac{1}{2r+1}}}{m} \vee 1 \leq l \leq \frac{n^{2r+1}}{m} \wedge (mn)^{\frac{1}{2r+1}}$. In this case $n \leq N_{ess} < n^{2r+1}$.

5. $N_{ess} = mn$ for $l \geq n \wedge (mn)^{\frac{1}{2r+1}}$.

We formulate the assumptions and give the detailed proof for upper and lower bounds in Sections 4 and 5 respectively. Combining the results of Theorems 4.2 and 5.3 directly implies Theorem 2.1. □

### 2.1. Phase Transitions from Spase to Dense Samples per Terminal

Within a logarithmic gap, the optimal rate $R(m, n, l, r)$ is fully characterized by $N_{ess}$, which can be written in the following equivalent form.

$$N_{ess} = \begin{cases} (2^l mn)^{\frac{2r+1}{2r+2}} \wedge lm \wedge mn, & \text{if } m \geq n^{2r+1}, \\ \left[ (lmn)^{\frac{2r+1}{2r+2}} \vee lm \right] \wedge mn, \\ \qquad \text{if } n^{2r} < m < n^{2r+1} \text{ and } n \leq m^{2r+1}, \\ (lmn)^{\frac{2r+1}{2r+2}} \wedge mn, \text{ if } m \leq n^{2r} \text{ and } n \leq m^{2r+1}, \\ (lm)^{2r+1} \wedge (lmn)^{\frac{2r+1}{2r+2}} \wedge mn, \\ \qquad \text{if } m \leq n^{2r} \text{ and } n > m^{2r+1}. \end{cases} \quad (3)$$

First, we see that the dependence of $R(m, n, l, r)$ on the communication budget $l$ is different for sparse and dense regimes. For the sparsest regime, i.e. $m \geq n^{2r+1}$, $R(m, n, l, r)$ first decays exponentially as $l$ increases, and then the rate slows to a polynomial order. For other regimes, $R(m, n, l, r)$ always depends polynomially on $l$.

Second, in order for the distributed system to achieve roughly the same performance as the centralized one, the minimum communication budget should be $n$ for $m > n^{2r}$ and $(mn)^{\frac{1}{2r+1}}$ for $m \leq n^{2r}$.

## 2.2. Comparisons with Related Previous Works

*Remark* 2.2. The work (Zaman & Szabó, 2022; Szabó & van Zanten, 2020) only considered the case $m \leq n^\gamma$ for $\gamma < 2r$ with an extra assumption $N_{ess} > n$. It is a special case of the last two cases in (3) by imposing that $N_{ess} > n$. In this case, $N_{ess} = (lmn)^{\frac{2r+1}{2r+2}} \wedge mn$ and Theorem 2.1 recovers the main results Theorems 1-2 of (Zaman & Szabó, 2022) under similar assumptions. Furthermore, our results extend those in (Zaman & Szabó, 2022) in two directions. First, we give the optimal rates for regimes $m < n^{2r}$, revealing more complicated but interesting structures of the problem. Second, we do not require $N_{ess} > n$. Although restricting the scope to $N_{ess} > n$ suffices in certain situations, it is generally impractical in most others. Moreover, investigating the problem without the restriction can give more insights from a theoretical view.

*Remark* 2.3. The work (Acharya et al., 2024) considered the density estimation problem with extremely sparse samples (i.e. $n = 1$), the $L^s$ norm and the Besov space $\mathcal{B}(p, q, r)$. By letting $p = q = s = 2$, the Besov space reduces to the Sobolev space $H^r([0,1], L)$ and the $L^s$ norm becomes the $L^2$ norm. In this way, their problem reduces to the special case of Example 2.6 with $n = 1$, in which their conclusion coincides with the first case in (3), that is $N_{ess} = (2^l m)^{\frac{2r+1}{2r+1}} \wedge m$. It is an interesting direction to generalize our results to the Besov space and $L^s$ norm, though it is not the main goal of this work due to space limitation.

*Remark* 2.4. We find the exponential dependence of $R(m, n, l, r)$ in $l$ for $m \geq n^{2r+1}$ and $1 \leq l \leq \frac{1}{2r+1} \log \frac{m}{n^{2r+1}}$. It is different from the Gaussian sequence model considered in (Zhu & Lafferty, 2018; Szabó & van Zanten, 2022; Cai & Wei, 2022) where $R(m, n, l, r)$ depends on $l$ only polynomially, although there are phase transitions in the polynomial order. This difference in minimax optimal rates is because the ways of generating samples are different, between the Gaussian sequence model in (Zhu & Lafferty, 2018; Szabó & van Zanten, 2022; Cai & Wei, 2022) and the random design problem in Section 1.3. For the former, the noisy versions of all the Fourier coefficients are known, instead of the random samples themselves for the latter.

*Remark* 2.5. In (Acharya et al., 2021) and (Yuan et al., 2024), the parametric distribution estimation problem was considered. The optimal rates were shown to be exponential in $l$ for some regimes, while polynomial for other regimes, which similar to the phenomenon revealed in this work. Deeper connections of these two problems are found and discussed in Remark 4.5.

## 2.3. Examples

In this subsection, consider several important estimation settings such as density estimation, Gaussian, binary, Pois-

son and heteroskedastic regression models. By directly verifying Assumption 4.1, 5.1 and 5.2 and specializing Theorem 2.1, the minimax optimal rates for all these settings are obtained in a unified manner.

### 2.3.1. NONPARAMETRIC DENSITY ESTIMATION

**Example 2.6** (Density Estimation)**.** Each of $m$ distributed terminals observes $n$ i.i.d. random samples $X_i^n$, and each sample $X_{ij}$ is generated from a density function $f \in H^r([0,1], L)$. We want to estimate $f$ at the decoder side. The problem is a special case of the framework in Section 1.3 by letting

$$p_f(x) = f(x), x \in [0, 1]. \tag{4}$$

The optimal rate for the above density estimation problem is shown as follows and proved in Appendix D.1.

**Corollary 2.7.** *For the nonparametric density estimation problem, we have $R(m, n, l, r) \preceq N_{ess}^{-\frac{2r}{2r+1}} \mathrm{Poly}(\log N)$ and $R(m, n, l, r) \succeq (N_{ess})^{-\frac{2r}{2r+1}} / \mathrm{Poly}(\log N)$.*

### 2.3.2. NONPARAMETRIC REGRESSION PROBLEMS

Consider nonparametric regression problems in the distributed setting. Each of $m$ distributed terminals observes $n$ i.i.d. pairs of random variables $(T_{ij}, Y_{ij})_{j=1}^n$, and each pair $(T_{ij}, Y_{ij})$ is sampled under random design, where we assume that the explanatory variable $T_{ij} \sim \mathrm{Unif}([0,1])$ and the response $Y_{ij}$ follows some distribution parameterized by $f(T_{ij})$. We want to estimate $f$ at the central decoder. Next we specify the conditional distribution $Y_{ij}|T_{ij}$ and consider the resulting regression problems.

**Example 2.8** (Nonparametric Gaussian Regression)**.** Let $Y_{ij}|T_{ij} \sim \mathcal{N}(f(T_{ij}), 1)$, and then the model is a special case of that in Section 1.3 by letting

$$p_f(t, y) = \frac{1}{\sqrt{2\pi}} e^{-\frac{(y-f(t))^2}{2}}, x = (t, y) \in [0, 1] \times \mathbb{R}. \tag{5}$$

**Example 2.9** (Nonparametric Binary Regression (Classification))**.** Let $f(t) \in [0, 1]$ for any $t \in [0, 1]$ and $Y_{ij}|T_{ij} \sim \mathrm{Bern}(f(T_{ij}))$. Then the model is a special case of that in Section 1.3 by letting

$$p_f(t, y) = f(t)\mathbb{1}_{y=1} + (1 - f(t))\mathbb{1}_{y=0}, \\ x = (t, y) \in [0, 1] \times \{0, 1\}. \tag{6}$$

**Example 2.10** (Nonparametric Poisson Regression)**.** Let $f(t) > 0$ for any $t \in [0, 1]$ and $Y_{ij}|T_{ij} \sim \mathrm{Poisson}(f(T_{ij}))$. Then the model is a special case of that in Section 1.3 by letting

$$p_f(t, y) = e^{-f(t)} \frac{(f(t))^y}{y!}, x = (t, y) \in [0, 1] \times \mathbb{N}. \tag{7}$$

**Example 2.11** (Nonparametric Heteroskedastic Regression). Let $f(t) > 0$ for any $t \in [0, 1]$ and $Y_{ij}|T_{ij} \sim \mathcal{N}(0, f(T_{ij}))$. Then the model is a special case of that in Section 1.3 by letting

$$p_f(t, y) = \frac{1}{\sqrt{2\pi f(t)}} e^{-\frac{y^2}{2f(t)}}, x = (t, y) \in [0, 1] \times \mathbb{R}. \quad (8)$$

For each of the above regression problems, the optimal rate is shown as follows and proved in Appendix D.2.

**Corollary 2.12.** *For the nonparametric Gaussion, binary, Poisson and heteroskedastic regression problems, we have*
$R(m, n, l, r) \preceq N_{ess}^{-\frac{2r}{2r+1}} \mathrm{Poly}(\log N)$ *and* $R(m, n, l, r) \succeq (N_{ess})^{-\frac{2r}{2r+1}} / \mathrm{Poly}(\log N)$.

## 3. Preliminary Results on Sobolev Spaces and Wavelets

We want to find good approximation formulas for the Sobolev space $H^r([0, 1], L)$, $r, L > 0$ in order to simplify the $L^2$ estimation problem in the space. It turns out that the wavelet construction is suitable, since it induces sparse representations of randomly generated data, which is fully exploited in the proof of the upper bound.

We follow the construction by (Daubechies, 1992), for more details see (Giné & Nickl, 2015; Simon, 1990). Start with two continuous father and mother wavelet functions $\phi$, $\psi$ with $S$ vanishing moments and bounded support on $[0, 2S - 1]$ and $[-S + 1, S]$ respectively, where $S > r$. By linear scaling of $\phi$ and $\psi$ and correcting them near the boundary, an orthonormal basis for $L^2[0, 1]$ is obtained.

We can construct an approximation $f^H \in H^r([0, 1], L)$ to $f$ in the $L^2$ norm with resolution $H \in \mathbb{N}$ by

$$f^H = \sum_{s=1}^{2^H} f_{Hs}\phi_{Hs}, \quad (9)$$

where $\{\phi_{Hs}\}_{s=1}^{2^H}$ is an orthonormal system and $f_{Hs} = (f, \phi_{Hs})$. Furthermore, for any function $f$ in the smaller space $H^r([0, 1], L) \subseteq L^2[0, 1]$, useful properties of the approximation $f^H$ are summarized in the following lemma. See Section 4.3 in (Giné & Nickl, 2015) and Corollary 26 in (Simon, 1990) for the proof.

**Lemma 3.1.** *Let* $f \in H^r([0, 1], L)$ *Then the approximation* $f^H$ *by* (9) *satisfies the following.*

1. $\|\phi_{Hs}\|_\infty \preceq 2^{\frac{H}{2}}$, *for any* $s = 1, ..., 2^H$.

2. *Let* $\mathcal{N}_{Hs} = \left\{ s' : [\frac{s-1}{2^H}, \frac{s}{2^H}] \cap \mathrm{supp}(\phi_{Hs'}) \neq \emptyset \right\}$ *for* $s = 1, ..., 2^H$. *Then* $|\mathcal{N}_{Hs}| \leq 2S + 2$.

3. *The* $L^2$ *convergence rate satisfies*

$$\|f - f^H\|_2^2 \preceq 2^{-2Hr}. \quad (10)$$

4. *If* $r > \frac{1}{2}$, *then* $H^r([0, 1], L) \subseteq L^\infty([0, 1], L')$ *for some* $L'(L, r) > 0$, *where* $L^\infty([0, 1], L') = \{f \in L^\infty([0, 1]) : \|f\|_{L^\infty} \leq L'\}$.

## 4. Upper Bounds

The wavelet-based approximation formula (9) in Section 3 is useful for estimating the function $f$. Throughout this section, let $K = 2^H$ to simplify the notations. In order to exploit (9), we make the following assumption regarding the existence of a good sample-wise estimator of each wavelet coefficient $f_{Hs}$.

**Assumption 4.1.** Assume $X = (T, Y)$, where $T \in [0, 1]$ and $Y \in \mathcal{Y}$. For any $H \in \mathbb{N}$ and $s = 1, ..., K$, there exists an estimator $\hat{f}_{Hs}(X) = h(Y)\phi_{Hs}(T)$ of $f_{Hs}$ that is unbiased ($\mathbb{E}[\hat{f}_{Hs}(X)] = f_{Hs}$) and sub-exponential with parameters $(\sqrt{c_1 K}, c_2 \sqrt{K})$.

The definitions and related properties of sub-exponential random variables can be found in Appendix A.1. In many estimation problems, such as the density estimation and regression problems in Section 2.3, the construction of the estimator $\hat{f}_{Hs}(X)$ in Assumption 4.1 is easily seen.

Then we can obtain the upper bound in the following theorem, which is the main goal of this section. The theorem is proved by the estimation protocol and its error analysis in the following two subsections respectively.

**Theorem 4.2.** *Under Assumption 4.1, we have* $R(m, n, l, r) \preceq (N_{ess})^{-\frac{2r}{2r+1}} \mathrm{Poly}(\log N)$.

### 4.1. The Layered Estimation Protocol

It suffices to let $l \leq n \wedge (mn)^{\frac{1}{2r+1}}$, otherwise we can simply discard the additional bits. That is, we consider Cases 1-4 in the proof of Theorem 2.1.

The main characteristic of our protocol is that it consists of two layers. The outer layer converts the original nonparametric distributed estimation problem into a distribution estimation problem. The inner layer estimates the parametric distribution. This can be achieved by invoking the protocol in Lemma A.2 for the distribution estimation problem. The resolution parameter $H$ of the wavelet approximation is carefully determined, so that the error induced by inner and outer layers is balanced.

**Preparation:** Choose the resolution parameters ($H, K = 2^H$) based on the parameters $(m, n, l, r)$, where $H$ is the

smallest integer such that

$$
2^{2S+2} \cdot K \geq
\begin{cases}
(2^l mn)^{\frac{1}{2r+2}}, & \text{for Case 1,} \\
(lm)^{\frac{1}{2r+1}}, & \text{for Case 2,} \\
\dfrac{lm}{2000 \log^2 N}, & \text{for Case 3,} \\
\dfrac{(lmn)^{\frac{1}{2r+2}}}{2000 \log^2 N}, & \text{for Case 4.}
\end{cases}
\tag{11}
$$

Let $K_0 = c_3 K^{\frac{1}{2}} \log N$, where $c_3$ is much larger than $c_2$ (e.g. $c_3 > 400(r+1)c_2$, cf. Assumption 4.1). Define the truncation function

$$
\text{Trunc}_{K_0}(w) = (w \wedge K_0) \vee (-K_0).
\tag{12}
$$

The truncation function and the sample-wise estimation function $\hat{f}_{Hs}$ are known to all the encoders and the decoder.

**Quantization:** For $i = 1, ..., m$, upon observing $X_i^n = (T_{ij}, Y_{ij})_{j=1}^n$, the $i$-th encoder first computes $S_{ij} = \lfloor KT_{ij} \rfloor$ and determines $\mathcal{N}_{HS_{ij}}$ based on $S_{ij}$. Then for each $j = 1, ..., n$ and $s \in \mathcal{N}_{HS_{ij}}$, the encoder computes $\hat{f}_{Hs}(X_{ij})$, $\tilde{f}_{Hs}(X_{ij}) = \text{Trunc}_{K_0}(\hat{f}_{Hs}(X_{ij}))$, and $Q_{Hs}(X_{ij}) = \frac{\hat{f}_{Hs}(X_{ij}) + K_0}{2K_0}$. It is easily seen that $Q_{Hs}(X_{ij}) \in [0, 1]$. Then it generates an i.i.d. random bit sequence $(V_{ij}(s))_{s \in \mathcal{N}_{HS_{ij}}}$ of length $|\mathcal{N}_{HS_{ij}}| = 2S+2$, and each bit follows the distribution $\text{Bern}(Q_{Hs}(X_{ij}))$. The sequence $V_{ij}$ is the quantization of $(\tilde{f}_{Hs}(X_{ij}))_{s \in \mathcal{N}_{HS_{ij}}}$.

Next, the $i$-th encoder computes $W_{ij} = (S_{ij}, V_{ij}) \in \{1, ..., K\} \times \{0, 1\}^{2S+2}$. Denote the alphabet of $W_{ij}$ by $\mathcal{W} = \{1, ..., K\} \times \{0, 1\}^{2S+2}$, then $|\mathcal{W}| = 2^{2S+2}K$. Note that $(W_{ij})_{i \in [1:m], j \in [1:n]}$ are i.i.d. random variables, and we denote the distribution of each $W_{ij}$ by $p_W(w)$. The $i$-th encoder holds $n$ i.i.d. samples $W_i^n = (W_{ij})_{j=1}^n$.

**Estimation of the Parametric Distribution $p_W$:** The encoders send messages to the decoder for estimation of the parametric distribution $p_W$ following the protocol $\text{ASR}(m, n, l, |\mathcal{W}|)$ introduced in Lemma A.2 and (Yuan et al., 2024). Let the estimate of the distribution be $\hat{p}_W(w)$.

**Decoding and Reconstruction:** Based on the estimate of the parametric distribution of $W_{ij}$, the decoder reconstructs an estimate $\bar{f}^H$ of $f$ as follows.

For any $s, s' = 1, ..., H$, the decoder computes

$$
\bar{f}_{Hs}^{(s')} =
\begin{cases}
2K_0 \left( \displaystyle\sum_{v: v(s)=1} \hat{p}_W(s', v) - \frac{1}{2}\hat{p}_W(s') \right), \\
\qquad\qquad\qquad\qquad \text{if } s \in \mathcal{N}_{Hs'}, \\
0, \qquad\qquad\qquad\quad \text{if } s \notin \mathcal{N}_{Hs'}.
\end{cases}
\tag{13}
$$

Then it computes

$$
\bar{f}_{Hs} = \sum_{s'} \bar{f}_{Hs}^{(s')},
\tag{14}
$$

and finally the estimate

$$
\bar{f}^H = \sum_{s=1}^K \bar{f}_{Hs} \phi_{Hs}.
\tag{15}
$$

*Remark* 4.3. The idea of estimating sparse wavelet coefficients through estimating a parametric distribution in nonparametric estimation problem is inspired by (Acharya et al., 2024), while only the density estimation problem with $n = 1$ was considered therein. The general estimation problem defined in Section 1.3 with $n > 1$ is substantially more difficult and require much more effort. There are two major differences.

First, the estimator $\hat{f}_{Hs}(X)$ is bounded for the density estimation problem. However, this is not the case for regression problems and the general estimation problem in Section 1.3. To overcome the difficulty, we truncate $\hat{f}_{Hs}(X)$ in the protocol and show the resulting error is negligible, under a sub-exponential condition of $\hat{f}_{Hs}(X)$ satisfied by most common problems.

Second, for the estimation of the parametric distribution, the work (Acharya et al., 2024) takes advantage of the simulate-and-infer protocol in (Acharya et al., 2020), which is optimal for $n = 1$ but not for $n > 1$. Unlike the special case $n = 1$, we can see from (Yuan et al., 2024) that the optimal protocols for the distribution estimation problem in Lemma A.2 with $n > 1$ vary across different parameter regimes. Hence there are many possible choices of the parameter $K$ and the protocol in the outer and inner layers respectively, and finding the optimal one can be obscure. The main work in this section is to determine the optimal parameter and protocol for cases 1-4 in the proof of Theorem 2.1, so that the optimal rate is always achieved.

### 4.2. Error Analysis

The following lemma describes the overall error bound in terms of the inner layer error. See Appendix B.1 for the proof.

**Lemma 4.4.** *For the estimate $\bar{f}^H$ obtained by the protocol in Section 4.1,*

$$
\mathbb{E}[\|\bar{f}^H - f\|_2^2] \preceq K^{-2r} + K \log^2 N \mathbb{E}\left[\|\hat{p}_W - p_W\|_2^2\right].
\tag{16}
$$

Next we bound the inner layer error for cases 1-4 by Lemma A.2 and (11). Then the overall error bounds are evaluated accordingly. We present the sketch here, and detailed verification can be found in Appendix B.2.

For Case 1, the condition of 3) in Lemma A.2 is satisfied and we have $\mathbb{E}[\|\hat{p}_W - p_W\|_2^2] \preceq \frac{2^{2S+2}\cdot K}{2^l mn}$. Then

$$\mathbb{E}[\|\bar{f}^H - f\|_2^2] \preceq (2^l mn)^{-\frac{r}{r+1}} \log^2 N.$$

For Case 2, the condition of 2) in Lemma A.2 is satisfied and $\mathbb{E}[\|\hat{p}_W - p_W\|_2^2] \preceq \frac{\log(\frac{2^{2S+2}\cdot K}{n}+1)}{ml} \vee \frac{1}{mn}$. Then

$$\mathbb{E}[\|\bar{f}^H - f\|_2^2] \preceq (lm)^{-\frac{2r}{2r+1}} \log^3 N.$$

For Case 3, the condition of 1) or 1') in Lemma A.2 is satisfied and roughly $\mathbb{E}[\|\hat{p}_W - p_W\|_2^2] \preceq \frac{2^{2S+2}\cdot K}{mnl} \vee \frac{1}{mn}$. Then

$$\mathbb{E}[\|\bar{f}^H - f\|_2^2] \preceq (lm)^{-2r} \log^{4r} N.$$

For Case 4, the condition of 1) in Lemma A.2 is satisfied, and we have $\mathbb{E}[\|\hat{p}_W - p_W\|_2^2] \preceq \frac{2^{2S+2}\cdot K}{mnl} \vee \frac{1}{mn}$. Then

$$\mathbb{E}[\|\bar{f}^H - f\|_2^2] \preceq (lmn)^{-\frac{r}{r+1}} \log^{4r+2} N.$$

Combining all cases completes the proof of Theorem 4.2.

*Remark* 4.5. From the construction of the protocol and its error analysis, we find a clear correspondence between the distribution estimation problem in Appendix A.3 and the nonparametric estimation problem considered in this work (cf. Section 1.3). With the help of the outer layer in our protocol, the protocol for the parametric distribution estimation can be used as an "oracle" prepared for the inner layer. The optimal rate for each case of the nonparamtric problem is implicitly achieved by a protocol for the distribution estimation problem.

From a high level, we can imagine a "homomorphism" from the distribution estimation problem to the nonparametric estimation problem, and each case of the former is mapped to one case of the latter. We hope this observation can give more insights to investigate various distributed statistical problems as a whole.

# 5. Lower Bounds

In this section, we establish the lower bounds and show the optimality of the estimation protocol.

Recall the wavelet construction in Section 3 and let $\psi_{hs}(t) = 2^{\frac{h}{2}}\psi(2^h t - s)$ for $h \in \mathbb{N}$ and $s \in \mathbb{Z}$. By the construction of $\psi$ in (Daubechies, 1992) we have $\|\psi_{hs}\|_2^2 = 1$. Let $h_0 = \lceil \log_2(2S+2) \rceil$ and $h \geq h_0$. For $s = 1, \cdots, 2^{h-h_0}$, there exists $s' \in \mathbb{Z}$ such that the support of $\psi_{h-h_0,s'}$ is contained in $\left[\frac{s-1}{2^{h-h_0}}, \frac{s}{2^{h-h_0}}\right]$. Let $\psi_s^{2^{h-h_0}} = \psi_{h-h_0,s'}$ for such $s'$. Then $\left\{\psi_s^{2^{h-h_0}}\right\}_{s=1}^{2^{h-h_0}}$ is a

subset of $\{\psi_{h-h_0,s'}\}_{s'\in\mathbb{Z}}$ such that the support of $\psi_s^{2^{h-h_0}}$ is contained in $\left[\frac{s-1}{2^{h-h_0}}, \frac{s}{2^{h-h_0}}\right]$.

Let $k = 2^{h-h_0}$, $\{\psi_s^k\}_{s=1}^k$ be a subset of $\{\psi_{h-h_0,s'}\}_{s'\in\mathbb{Z}}$, where the support of $\psi_s^k$ is contained in $\left[\frac{s-1}{k}, \frac{s}{k}\right]$ discussed above. To construct multiple hypotheses that are useful for the proof, consider a finite sieve $\mathcal{F}(k, C_0, \epsilon)$ defined as

$$\left\{ f_{z^k} : f_{z^k} = C_0 + \epsilon k^{-(r+\frac{1}{2})} \sum_{s=1}^k z_s \psi_s^k, z^k \in \{-1, 1\}^k \right\}. \tag{17}$$

The constant $C_0$ depends on the problem. Specifically, it is 1 for the density estimation problem, 0 for Gaussian regression, $\frac{1}{2}$ for classification and a positive real number for Poisson and heteroskedastic regression in Section 2.3. Let $\epsilon \in (0, 1)$ be small enough such that, i) $\mathcal{F}(k, C_0, \epsilon) \subseteq H^r([0,1], L)$; ii) if $C_0 > 0$, then $f_{z^k}(t) \in [\frac{C_0}{2}, \frac{3C_0}{2}], \forall t \in [0, 1]$. This can be achieved since we have $\|\psi_s^k\|_{H^r} \preceq 2^{hr}$ and $\|\psi_s^k\|_\infty \preceq 2^{\frac{h}{2}}$. To simplify the notation, let $p_{z^k} = p_{f_{z^k}}$ for any function $f_{z^k} \in \mathcal{F}(k, C_0, \epsilon)$ and correspondingly, $\mathbb{P}_{z^k} = \mathbb{P}_{X^n \sim p_{z^k}^n}$ and $\mathbb{E}_{z^k} = \mathbb{E}_{X^n \sim p_{z^k}^n}$, where the meaning will be clear in the context.

Then we make a few assumptions on the sample distribution $p_{z^k}$, which are essential to prove the lower bounds.

**Assumption 5.1.** Assume that $X = (T, Y)$, where $T \in [0, 1]$ and $Y \in \mathcal{Y}$. For any $z^k \in \{\pm 1\}^k$ and $s = 1, ..., k$, $p_{z^k}(x \in [\frac{s-1}{k}, \frac{s}{k}] \times \mathcal{Y}) = \frac{1}{k}$ and the conditional distribution $p_{z^k}(x | x \in [\frac{s-1}{k}, \frac{s}{k}] \times \mathcal{Y})$ only depends on $z_s$. Specifically, the distribution $p_{z^k}$ admits a decomposition

$$p_{z^k}(x) = \frac{1}{k} p_{s, z_s}(x). \tag{18}$$

for any $x = (t, y)$ with $t \in [\frac{s-1}{k}, \frac{s}{k}]$, where $p_{s, z_s}(x)$ is a distribution on $[\frac{s-1}{k}, \frac{s}{k}] \times \mathcal{Y}$ for any $s = 1, ..., k$ and $z_s = \pm 1$.

**Assumption 5.2.** Let the sample-wise log-likelihood ratio be

$$L_{s, z_s}(x) \triangleq \log \left( \frac{p_{s, -z_s}(x)}{p_{s, z_s}(x)} \right). \tag{19}$$

for any $x = (t, y)$ with $t \in [\frac{s-1}{k}, \frac{s}{k}]$, $s = 1, ..., k$ and $z_s = \pm 1$. We assume that for $X \sim p_{s, z_s}$, $L_{s, z_s}(X)$ is sub-exponential with parameters $(\nu = \sqrt{C_1 \cdot k^{-2r}}, \beta = C_2 k^{-r})$ and $|\mathbb{E}[L_{s, z_s}(X)]| \leq C_3 \cdot k^{-2r}$.

Then we have the main theorem of this section, focusing on the lower bound. It is proved in the rest of this section.

**Theorem 5.3.** *Under Assumptions 5.1 and 5.2, we have* $R(m, n, l, r) \succeq (N_{ess} \cdot \text{Poly}(\log N))^{-\frac{2r}{2r+1}}$.

## 5.1. Information-Theoretic Lower Bounding Methods

We define a prior distribution on $H^r([0,1], L)$ to be the uniform distribution on the sieve $\mathcal{F}(k, C_0, \epsilon)$. Let $Z^k =$

$\{Z_s\}_{s=1}^k$ be a sequence of i.i.d. Rademacher random variables with mean 0. Then under the prior distribution, the function and the sample distribution are $f = f_{Z^k}$ and $p_f = p_{Z^k}$, respectively. Then we have the following lemma proved in Appendix C.1.

**Lemma 5.4.** *If $\frac{1}{k}\sum_{s=1}^k I(Z_s; B^m) \leq \frac{1}{2}$ for some $k \in \mathbb{N}$, then $R(m, n, l, r) \succeq k^{-2r}$.*

With the help of Lemma 5.4, the proof of Theorem 5.3 is reduced to choosing suitable $k$ and showing the information inequality $\frac{1}{k}\sum_{s=1}^k I(Z_s; B^m) \leq \frac{1}{2}$, and then we obtain that $R(m, n, l, r) \succeq k^{-2r}$. Methods to prove the inequality are different for Cases 1-5. The bounds for Case 5 and Case 3 are easy and shown in Appendices C.2 and C.3. The proof for the other three cases need much more efforts, which is the goal of the remaining parts of this section.

## 5.2. Proof for the Remaining Cases

First we define the terminal-wise likelihood ratio to be

$$\mathcal{L}_{s,z^k}(x^n) \triangleq \frac{p_{z^k \odot e_s}^n(x^n)}{p_{z^k}^n(x^n)}, \qquad (20)$$

where $z^k \odot z'^k = (z_{s'} \cdot z'_{s'})_{s'=1}^k$ and $e_s = ((-1)^{\mathbb{1}_{s'=s}})_{s'=1}^k$. It plays a central role in the rest of the proof, since two of its properties lead to different kinds of bounds for $\frac{1}{k}\sum_{s=1}^k I(Z_s; B^m)$. The following two lemmas describe the bounds respectively, whose detailed proof can be found in Appendix C.4.

**Lemma 5.5.** *If $\mathbb{E}_{z^k}[(\mathcal{L}_{s,z^k}(X^n) - 1)^2] \leq \alpha^2$ for any $s = 1, ..., k$ and $z^k \in \{-1, 1\}^k$, then we have*

$$\frac{1}{k}\sum_{s=1}^k I(Z_s; B^m) \leq \frac{2^l m \alpha^2}{2k}. \qquad (21)$$

**Lemma 5.6.** *If there exists a Boolean function $E(x^n)$ such that*
*i) $\mathbb{P}_{z^k}[E(X^n) = 0] \leq \delta_1 < \frac{1}{2}$ for any $z^k \in \{-1, 1\}^k$;*
*ii) $E(x^n) = 1$ if and only if $|\mathcal{L}_{s,z^k}(x^n) - 1| \leq \delta_2$ for any $z^k \in \{-1, 1\}^k$, $s = 1, ..., k$.*
*Then we have*

$$\frac{1}{k}\sum_{s=1}^k I(Z_s; B^m) \leq m((\log 2)\delta_1^{\frac{1}{2}} + \delta_1) + \frac{16ml\delta_2^2}{k}. \qquad (22)$$

*Remark* 5.7. The idea behind Lemma 5.6 and its proof is similar to Theorem 2 in (Acharya et al., 2023), that is to obtain a tight strong data processing constant. The major improvements of Lemma 5.6 is that it admits an irregular event $\{E(X^n) = 0\}$ with small probability, while such an event is not included in Theorem 2 in (Acharya et al., 2023). This makes Lemma 5.6 more flexible to use especially for various regression problems (e.g. the nonparametric Gaussian regression problem in Example 2.8), since

in these problems sub-Gaussian or boundedness condition of the terminal-wise likelihood ratio $\mathcal{L}_{s,z^k}(X^n)$ is not satisfied. To overcome the difficulty, we establish the bound of $\mathcal{L}_{s,z^k}(X^n)$ on the regular event $\{E(X^n) = 1\}$ and then argue that the effect of the irregular event $\{E(X^n) = 1\}$ with small probability is limited in Lemma 5.5.

*Remark* 5.8. In the work (Cai & Wei, 2022), similar bound (18) therein for information measures is simpler and makes further analysis easier. However, the proof there depends heavily on the symmetry of the Gaussian random variable, which clearly does not hold for the general estimation problems in this work. Hence we have to handle the small error probability more carefully. This also explains the reason that the work (Cai & Wei, 2022) obtains a tight bound for the Gaussian sequence model, but it is much harder to eliminate the polynomial factor of $\log N$ for the problem here.

Then we specialize Lemmas 5.5 and 5.6 to Cases 1, 2 and 4 and derive the corresponding bounds. The goal is to bound $\mathcal{L}_{s,z^k}(X^n) - 1$ itself or its second moment. To achieve the goal, the underlying intuition is described as follows. By Assumption 5.1, the terminal-wise likelihood ratio $\mathcal{L}_{s,z^k}(x^n)$ is related to the sample-wise one in (19) by

$$\mathcal{L}_{s,z^k}(x^n) = \prod_{j:t_j \in [\frac{s-1}{k}, \frac{s}{k}]} \exp\left(L_{s,z_s}(x_j)\right), \qquad (23)$$

Hence $\mathcal{L}_{s,z^k}(X^n)$ is a product of many independent factors $\exp(L_{s,z_s}(X_j))$, if $(T_i)_{i=1}^n$ is given and $X_i = (T_i, Y_i)$. By Assumption 5.2, each of these factors has a small amplitude. If the number of these factors is bounded, then both $\mathcal{L}_{s,z^k}(X^n) - 1$ and its second moment can be bounded. The number has a clear meaning by Assumption 5.1. It is the number of balls in the $s$-th bin (denoted by $V_s$) in the classic balls and bins model where $n$ balls are thrown into $k$ bins at random (see Appendix A.4 for details). By bounding $V_s$, the goal can be achieved and the whole proof is completed. We sketch the proof for each case in the following, and details can be found in Appendices C.5, C.6 and C.7 respectively.

### 5.2.1. PROOF FOR CASE 1

The goal is to show $R(m, n, l, r) \succeq (2^l mn)^{-\frac{r}{r+1}}$ for Case 1. Note that the expectation of each $\exp(2L_{s,z_s}(X))$ is roughly $\exp(2\nu^2) \approx 1 + 2\nu^2$ by Assumption 5.2 and the number $V_s$ is $\frac{n}{k}$ on average. Then by (23), for $k \asymp (mn2^l)^{\frac{1}{2r+2}}$, we can show that

$$\mathbb{E}_{z^k}[(\mathcal{L}_{s,z^k}(X^n) - 1)^2] = O\left(\frac{n}{k}\nu^2\right).$$

Hence Lemma 5.5 implies that

$$\frac{1}{k}\sum_{s=1}^k I(Z_s; B^m) = O\left(\frac{2^l mn}{k^{2r+2}}\right).$$

Finally, by Lemma 5.4 we complete the proof.

### 5.2.2. PROOF FOR CASE 2

We need to show $R(m,n,l,r) \succeq (lm \log^2 m)^{-\frac{2r}{2r+1}}$ for Case 2. With the choice $k \asymp (ml \log^2 m)^{\frac{1}{2r+1}}$, we want to obtain a bound for $\mathcal{L}_{s,z^k}(X^n) - 1$ with a large probability. Instead, we turn to bound $\log \mathcal{L}_{s,z^k}(X^n)$. By Lemma A.3, we obtain a uniform bound for all these numbers $V_s$,

$$\max_{1 \le s \le k} V_s \preceq \log m$$

with probability $1 - m^{-100}$. Based on this event, by (23) and Assumption 5.2 we can further show that

$$|\log \mathcal{L}_{s,z^k}(X^n)| \preceq \beta \log m$$

with probability $1 - m^{-100}$. Thus we can choose $\delta_1 = O(m^{-100})$ and $\delta_2 = O(\beta \log m)$ in Lemma 5.6, which implies that

$$\frac{1}{k} \sum_{s=1}^{k} I(Z_s; B^m) = O\left(\frac{ml \log^2 m}{k^{2r+1}}\right).$$

Finally, by Lemma 5.4 we complete the proof.

### 5.2.3. PROOF FOR CASE 4

We need to show $R(m,n,l,r) \succeq (lmn)^{-\frac{r}{r+1}} \log^{-(2r+3)} n$ for Case 4. With the choice $k \asymp (mnl)^{\frac{1}{2r+2}} \log^{\frac{2r+3}{2r+1}} n$, we aim at bounding $\log \mathcal{L}_{s,z^k}(X^n)$ similar to the previous case. By Lemma A.3, we have

$$\max_{1 \le s \le k} V_s \preceq \frac{n \log^3 n}{k}$$

with probability $1 - n^{-100(r+1)}$. Based on this event, by (23) and Assumption 5.2 we can further show that

$$|\log \mathcal{L}_{s,z^k}(X^n)| \preceq \sqrt{\frac{\nu^2 n \log^4 n}{k}}$$

with probability $1 - n^{-100(r+1)}$. Thus we can choose $\delta_1 = O(n^{-100(r+1)})$ and $\delta_2 = O\left(\sqrt{\frac{\nu^2 n \log^4 n}{k}}\right)$ in Lemma 5.6, which implies that

$$\frac{1}{k} \sum_{s=1}^{k} I(Z_s; B^m) = O\left(\frac{mnl \log^4 n}{k^{2r+2}}\right).$$

Finally, by Lemma 5.4 we complete the proof.

## 6. Conclusion and Discussions

In this work, we considered the distributed nonparametric function estimation problems under communication constraints. Specifically, we quantitatively characterized the optimal rates as the number of samples per terminal varies from sparse to dense, finding interesting phase transitions.

Our work motivates several future directions. First, our framework focused on the estimation of functions in Sobolev type spaces with $L^2$ norm. Considering general Besov spaces and the $L^q$ norm may lead to further findings. Second, the differential privacy is another important consideration in distributed estimation or federated learning. How the optimal rates depend on the sample size per terminal under privacy constraints is also an interesting problem, dual to the problem under communication constraints explored in this work. Third, in many practical settings the Sobelov regularity parameter $r$ may be unknown. Adaptive estimation protocols for $r$ has been considered for several specific settings (e.g. (Szabó & van Zanten, 2020; Cai & Wei, 2022)), and may be useful for our problem. Moreover, it is interesting to develop adaptive protocols for other parameters like $(m,n)$, which is studied in the distributed kernel ridge problem by (Zhang et al., 2015; Xu et al., 2018; 2019). Finally, we remark that problems with different ways of sample generation may need separate study. We concentrate on the random design setting in this work. Our methods may be useful for the analysis of the fixed design setting (Li et al., 2024), while the optimal rates may be different. In Remark 2.4 we compared the difference of our problem with the Gaussian sequence model. More generally, heterogeneous or non i.i.d. data for different terminals raise more interesting problems that have both theoretical and practical meaning.

## Acknowledgements

Deheng Yuan would like to thank Professor Yuhong Yang for helpful discussions, which brought the communication-constrained estimation problems to his attention.

This work was supported in part by the NSFC Projects No.12025104 and 62301144, in part by the SEU Startup Fund No.RF1028623030, and in part by the Zhishan Young Scholar Fund No.2242025RCB0032.

## Impact Statement

This paper presents work whose goal is to advance the field of Machine Learning. There are many potential societal consequences of our work, none of which we feel must be specifically highlighted here.

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

# A. Preliminary Definitions and Results

## A.1. Sub-Gaussian and Sub-Exponential Random Variables

We give definitions and properties of sub-Gaussian and sub-exponential random variables that are useful for this work. More details can be found in (Wainwright, 2019).

A random variable $X$ with mean $\mu = \mathbb{E}[X]$ is sub-exponential with parameters $(\nu, \beta)$ if

$$\mathbb{E}[e^{\lambda(X-\mu)}] \leq e^{\frac{\nu^2 \lambda^2}{2}}, \forall |\lambda| < \frac{1}{\beta}.$$

Note that a sub-exponemtial random has finite moments of any order.

A random variable $X$ is sub-Gaussian with parameter $\sigma$ if it is subexponential with parameter $(\sigma, 0)$, where $\frac{1}{0}$ is interpreted as $\infty$. Then a random vector $X^n$ is called sub-Gaussian with parameter $\sigma$ if for any unit vector $v^n$, $\sum_{j=1}^{n} v_j X_j$ is sub-Gaussian with parameter $\sigma$.

If $X \in [a, b]$, then it is sub-Gaussian with parameter $\frac{b-a}{2}$. See the discussion after Proposition 2.5 in (Wainwright, 2019) for the proof. Let $X^n$ be a random vector, if $X_j$ for $j = 1, ..., n$ are independent and each $X_j$ is a sub-Gaussian random variable with parameter $\sigma$, then it is easy to verify that $X^n$ is a sub-Gaussian random vector with parameter $\sigma$.

Let $X_j$, $j = 1, ..., n$ be i.i.d. random variables and $X_j$ is sub-exponential with parameters $(\nu_j, b_j)$. Then we can verify that $\sum_{j=1}^{n} a_j X_j$ is sub-exponential with parameters $(\sqrt{\sum_{j=1}^{n} a_j^2 \nu_j^2}, \max_{1 \leq j \leq n} b_j)$. The following lemma characterizes the tail bound for a sub-exponential random variable, which is by Proposition 2.9 in (Wainwright, 2019).

**Lemma A.1.** *Let $X$ be a sub-exponential random variable with mean $\mu$ and parameters $(\nu, \beta)$. Then*

$$\mathbb{P}[|X - \mu| \geq t] \leq \begin{cases} 2 \exp\left(-\frac{t^2}{2\nu^2}\right), & 0 \leq t \leq \frac{\nu^2}{\beta}, \\ 2 \exp\left(-\frac{t}{2\beta}\right), & t > \frac{\nu^2}{\beta}. \end{cases} \tag{24}$$

## A.2. Preliminaries on divergences between distributions

Let $p_1(u)$ and $p_2(u)$ be two distributions over $\mathcal{U}$, the KL divergence is defined by

$$D_{\mathcal{U}}(p_1(u)||p_2(u)) = \mathbb{E}_{p_1}\left[\log \frac{p_1(U)}{p_2(U)}\right].$$

The $\chi^2$ divergence is defined by

$$\chi_{\mathcal{U}}^2(p_1(u)||p_2(u)) = \mathbb{E}_{p_2}\left[\left(\frac{p_1(U)}{p_2(U)} - 1\right)^2\right].$$

By the convexity of the logarithm function, it is easy to see that

$$D_{\mathcal{U}}(p_1(u)||p_2(u)) \leq \chi_{\mathcal{U}}^2(p_1(u)||p_2(u)). \tag{25}$$

## A.3. Protocols for Estimating a Parametric Distribution

Suppose that we want to estimate a parametric distribution $p_W$ over a finite set $\mathcal{W}$ with size $k = |\mathcal{W}|$. The setting is the same as Section 1.3, except that the task is different. To be precise, there are $m$ encoders and the $i$-th encoder holds $n$ i.i.d. samples $(W_{ij})_{j=1}^{n}$. Each encoder can send a length $l$ message to help the decoder establish an estimate $\hat{p}_W(w)$, such that the $L^2$ loss $\mathbb{E}[\|\hat{p}_W - p_W\|_2^2]$ is minimized.

The following lemma is essential, which characterizes the optimal error rates (up to logarithmic factors, cf. (Yuan et al., 2024)) for the above distribution estimation problem.

**Lemma A.2.** *For the above distribution estimation problem, there exists an interactive protocol $\mathrm{ASR}(m, n, l, k)$ such that for any $\boldsymbol{p}_W \in \Delta_{\mathcal{W}}$, the protocol outputs an estimate $\hat{\boldsymbol{p}}_W$ satisfying,*

*1) if $k \leq n$, $m(l \wedge k) > 1000k \log^2 N$, then $\mathbb{E}[\|\hat{\boldsymbol{p}}_W - \boldsymbol{p}_W\|_2^2] = O\left(\frac{k}{mnl} \vee \frac{1}{mn}\right)$;*

*1') if $k \leq n$, $l \geq \log n$ and $m\lfloor \frac{l}{\lceil \log n \rceil} \rfloor \geq k$, then $\mathbb{E}[\|\hat{\boldsymbol{p}}_W - \boldsymbol{p}_W\|_2^2] = O\left(\frac{k \log n}{mnl} \vee \frac{1}{mn}\right)$;*

*2) if $n < k \leq (2^l - 1) \cdot n$, $l \geq 2$ and $m(l \wedge n) > 2000n \log^2 N$, then $\mathbb{E}[\|\hat{\boldsymbol{p}}_W - \boldsymbol{p}_W\|_2^2] = O\left(\frac{\log(\frac{k}{n}+1)}{ml} \vee \frac{1}{mn}\right)$;*

*3) if $k > (2^l - 1) \cdot n$, $l \geq 4$ and $m(l \wedge n) > 4000n \log^2 N$, then $\mathbb{E}[\|\hat{\boldsymbol{p}}_W - \boldsymbol{p}_W\|_2^2] = O\left(\frac{k}{2^l mn}\right)$.*

The cases in 1-3) are achieved by (Yuan et al., 2024), see Theorem 1 therein for the proof. The proof of 1') can be found in Appendix A.3.1.

### A.3.1. PROOF OF LEMMA A.2

It remains to show the case 1').

**The Estimation Protocol**

Let $l' = \lfloor \frac{l}{\lceil \log n \rceil} \rfloor \wedge k$ and $m' = \lfloor \frac{ml'}{k} \rfloor$. Each encoder divides its $l$ bits into $l'$ frames, and there are $\lceil \log n \rceil$ bits in each frame. Then for each $w \in \mathcal{W}$, each frame is sufficient for encoding the number of $w$ among the $n$ samples at each encoder. We can allocate $m'k$ frames to all the $w \in \mathcal{W}$, such that the frames held by the same encoder are allocated to different $w$, and exactly $m'$ frames are allocated to each $w \in \mathcal{W}$.

Each encoder then encodes the number of $w$ among its $n$ samples to each frame, where the frame is allocated to $w \in \mathcal{W}$. Then it connects all its frames and sends them to the decoder. For each $w \in \mathcal{W}$, the decoder computes $num(w)$ by summing up the number of $w$, where each number is encoded in one of the $m'$ frames allocated to $w$. Then it computes $\hat{p}_W(w) = \frac{num(w)}{m'n}$ and outputs the estimate $\hat{\boldsymbol{p}}_W$.

**Error Analysis**

For each $w \in \mathcal{W}$ it is easy to see that $num(w)$ is the sum of $m'n$ i.i.d. random variables, and each of them follows the distribution $\text{Bern}(p_W(w))$. So we have $\mathbb{E}[|\hat{p}_W(w) - p_W(w)|^2] = O(\frac{p_W(w)(1-p_W(w))}{m'n})$ and $\mathbb{E}[\|\hat{p}_W(w) - p_W(w)\|_2^2] = O(\frac{1}{m'n}) = O(\frac{k \log n}{mnl} \vee \frac{1}{mn})$, completing the proof.

### A.4. Analysis of the Balls and Bins Model

Suppose there are $n$ balls and $k$ bins and each ball is independently put into a bin at random. Let $V_s$ be the number of balls in the $s$-th bin. Then it is a sum of $n$ i.i.d. $\text{Bern}(\frac{1}{k})$ random variables. We are interested in the maximal number of balls over all $k$ bins, namely $\max_{1 \leq s \leq k} V_s$, which is related to the strong data processing constant in Section 5. We can obtain the following inequalities characterizing cases $k \geq n$ and $n \geq k$ respectively.

**Lemma A.3.** *1. Let $k \geq n$ and $c \in \mathbb{N}$ be sufficiently large, then*

$$\mathbb{P}\left[\max_{1 \leq s \leq k} V_s \geq c + 1\right] \leq k \exp\left(-\frac{c}{2}\right). \tag{26}$$

*2. Let $n \geq k$ and $c \in \mathbb{N}$ be sufficiently large, then*

$$\mathbb{P}\left[\max_{1 \leq s \leq k} V_s \geq \frac{cn}{k}, \forall s = 1, ..., k\right] \leq k \exp\left(-\frac{cn}{8k}\right). \tag{27}$$

*Proof.* By the Chernoff's bound, we have

$$\mathbb{P}\left[V_s \geq (1 + c')\frac{n}{k}\right] \leq \exp\left(-\frac{c'^2 n}{(2 + c')k}\right), \forall c' > 0. \tag{28}$$

Then we can derive the desired inequalities as follows.

**Proof of Case 1:** By letting $c' = \frac{k}{n}(1 + c) - 1$ in (28), we have

$$\mathbb{P}[V_s \geq c + 1] \leq \exp\left(-\frac{\left[\frac{k}{n}(1 + c) - 1\right]^2 n}{\left[\frac{k}{n}(1 + c) + 1\right] k}\right)$$

$$\leq \exp\left(-\frac{\left(\frac{k}{n}c\right)^2 n}{\left(\frac{2k}{n}c\right) k}\right) = \exp\left(-\frac{c}{2}\right).$$

By applying the union bound, we complete the proof.

**Proof of Case 2:** By letting $c' = c - 1$ in (28), we have

$$\mathbb{P}\left[V_s \geq \frac{cn}{k}\right] \leq \exp\left(-\frac{(c - 1)^2 n}{(c + 1)k}\right)$$

$$\leq \exp\left(-\frac{(\frac{c}{2})^2 n}{2ck}\right) = \exp\left(-\frac{cn}{8k}\right).$$

The conclusion is then implied by the union bound.

$\square$

## B. Error Analysis for the Protocol in Section 4

### B.1. Proof of Lemma 4.4

The overall error can be bounded as follows.

$$\mathbb{E}[\|\bar{f}^H - f\|_2^2] \leq 2\left(\mathbb{E}[\|\bar{f}^H - f^H\|_2^2] + \mathbb{E}[\|f^H - f\|_2^2]\right)$$

$$\preceq \sum_{s=1}^{K} \mathbb{E}[|f_{Hs} - \bar{f}_{Hs}|^2] + 2^{-2Hr}$$

$$= \sum_{s=1}^{K} \mathbb{E}[|f_{Hs} - \bar{f}_{Hs}|^2] + K^{-2r},$$

where the second inequality is because $(\phi_{Hs})_{s=1}^{K}$ is an orthonormal system and (10) in Lemma 3.1 holds, and the last equality is by $K = 2^H$. Then it suffices to bound the first term $\sum_{s=1}^{K} \mathbb{E}[|f_{Hs} - \bar{f}_{Hs}|^2]$. By the bias-variance decomposition, we have

$$\sum_{s=1}^{K} \mathbb{E}[|f_{Hs} - \bar{f}_{Hs}|^2] \leq \sum_{s=1}^{K} \left|f_{Hs} - \mathbb{E}\left[\tilde{f}_{Hs}(X)\right]\right|^2 + \sum_{s=1}^{K} \mathbb{E}\left[\left|\bar{f}_{Hs} - \mathbb{E}\left[\tilde{f}_{Hs}(X)\right]\right|^2\right]. \tag{29}$$

It suffices to bound two terms on the right hand side of (29) respectively.

**The first term in** (29)**:** Note that

$$\left|f_{Hs} - \mathbb{E}\left[\tilde{f}_{Hs}(X)\right]\right| = \left|\mathbb{E}\left[\hat{f}_{Hs}(X) - \tilde{f}_{Hs}(X)\right]\right|$$

$$\leq \mathbb{E}\left[\mathbb{1}_{|\hat{f}_{Hs}(X)| > K_0}\left(\left|\hat{f}_{Hs}(X)\right| - K_0\right)\right]$$

$$= \int_0^\infty \mathbb{P}\left[|\hat{f}_{Hs}(X)| > K_0 + a\right] da$$

By Assumption 4.1, $\hat{f}_{Hs}(X)$ is sub-exponential with parameters $(\sqrt{c_1 K}, c_2\sqrt{K})$. Hence $\hat{f}_{Hs}(X)$ has finite second moment, i.e., for some $\Sigma > 0$,

$$\mathbb{E}[|\hat{f}_{Hs}(X)|^2] \leq \Sigma^2. \tag{30}$$

Note that

$$K_0 = c_3 K^{\frac{1}{2}} \log N \asymp K^{\frac{1}{2}} \log N \succeq K^{\frac{1}{2}} \asymp \frac{c_1 K}{c_2\sqrt{K}}$$

and by (30),

$$\mathbb{E}[|\hat{f}_{Hs}(X)|] \le \sqrt{\mathbb{E}[|\hat{f}_{Hs}(X)|^2]} \le \Sigma = O(1).$$

Then by Lemma A.1 we have

$$\int_0^\infty \mathbb{P}\left[|\hat{f}_{Hs}(X)| > K_0 + a\right] da$$

$$\le \int_0^\infty 2\exp\left(-\frac{K_0 + a}{4c_2 K^{\frac{1}{2}}}\right) da \asymp K^{\frac{1}{2}}\exp\left(-\frac{K_0}{4c_2 K^{\frac{1}{2}}}\right) \preceq \sqrt{K}N^{-100},$$

where the last step is by the choice of $c_3$ with $c_3 > 400(r+1)c_2$, and then $K_0 = c_3 K^{\frac{1}{2}}\log N \ge 100(r+1)\log N \cdot (4c_2 K^{\frac{1}{2}})$. Then we have

$$\sum_{s=1}^K \left|f_{Hs} - \mathbb{E}\left[\tilde{f}_{Hs}(X)\right]\right|^2 \preceq K^{\frac{3}{2}}N^{-100(r+1)} \preceq K^{-2r}. \tag{31}$$

since we have $K \preceq N$ by the choice of $K$ in (11).

**The second term in** (29)**:** Since $V(s)|X \sim \text{Bern}(Q_{Hs}(X))$ and $Q_{Hs}(X) = \frac{\tilde{f}_{Hs}(X)+K_0}{2K_0}$, then

$$\sum_{s':s\in\mathcal{N}_{Hs'}} 2K_0\left(\sum_{v:v(s)=1} p_W(s',v) - \frac{1}{2}p_W(s')\right) = \sum_{s':s\in\mathcal{N}_{Hs'}} 2K_0\left(\mathbb{E}\left[\mathbb{1}_{S=s'}\left(\sum_{v:v(s)=1}\mathbb{P}[V=v|X] - \frac{1}{2}\right)\right]\right)$$

$$= \sum_{s':s\in\mathcal{N}_{Hs'}} 2K_0\left(\mathbb{E}\left[\mathbb{1}_{S=s'}\left(\mathbb{P}[V(s)=1|X] - \frac{1}{2}\right)\right]\right)$$

$$= \sum_{s':s\in\mathcal{N}_{Hs'}}\left(\mathbb{E}\left[\mathbb{1}_{S=s'} \cdot 2K_0\left(Q_{Hs}(X) - \frac{1}{2}\right)\right]\right)$$

$$= \sum_{s':s\in\mathcal{N}_{Hs'}}\left(\mathbb{E}\left[\mathbb{1}_{S=s'}\tilde{f}_{Hs}(X)\right]\right)$$

$$= \mathbb{E}\left[\mathbb{1}_{s\in\mathcal{N}_{HS}}\tilde{f}_{Hs}(X)\right]$$

$$= \mathbb{E}\left[\tilde{f}_{Hs}(X)\right].$$

where the last equality is because $\tilde{f}_{Hs}(X) = \hat{f}_{Hs}(X) = 0$ for $s \notin \mathcal{N}_{HS}$. Then by (13) and (14), we have

$$\left|\bar{f}_{Hs} - \mathbb{E}\left[\tilde{f}_{Hs}(X)\right]\right|^2$$

$$= \left|\sum_{s':s\in\mathcal{N}_{Hs'}} 2K_0\left(\sum_{v:v(s)=1}\hat{p}_W(s',v) - \frac{1}{2}\hat{p}_W(s')\right) - \sum_{s':s\in\mathcal{N}_{Hs'}} 2K_0\left(\sum_{v:v(s)=1} p_W(s',v) - \frac{1}{2}p_W(s')\right)\right|^2$$

$$= K_0^2\left|\sum_{s':s\in\mathcal{N}_{Hs'}}\left(\sum_{v:v(s)=1}(\hat{p}_W(s',v) - p_W(s',v)) + \sum_{v:v(s)=0}(p_W(s',v) - \hat{p}_W(s',v))\right)\right|^2$$

$$\le K_0^2(2S+2)2^{2S+2}\sum_{s':s\in\mathcal{N}_{Hs'}}\sum_v |\hat{p}_W(s',v) - p_W(s',v)|^2,$$

where the last step is by the Cauchy-Schwarz inequality and $|\mathcal{N}_{Hs'}| \leq 2S+2$. By taking the summation, we have

$$\sum_{s=1}^{K} \left| \bar{f}_{Hs} - \mathbb{E}\left[\tilde{f}_{Hs}(X)\right] \right|^2$$

$$\leq K_0^2(2S+2)2^{2S+2} \sum_{s'=1}^{K} \sum_{s:s\in\mathcal{N}_{Hs'}} \sum_{v} |\hat{p}_W(s',v) - p_W(s',v)|^2$$

$$\leq K_0^2(2S+2)^2 2^{2S+2} \sum_{s'=1}^{K} \sum_{v} |\hat{p}_W(s',v) - p_W(s',v)|^2.$$

Taking the expectation, we have

$$\sum_{s=1}^{K} \mathbb{E}\left[\left|\bar{f}_{Hs} - \mathbb{E}\left[\tilde{f}_{Hs}(X)\right]\right|^2\right] \preceq K \log^2 N \|\hat{p}_W - p_W\|_2^2. \tag{32}$$

**Completing the proof:** Combining (29), (31) and (32), we have

$$\sum_{s=1}^{K} \mathbb{E}[|f_{Hs} - \bar{f}_{Hs}|^2] \preceq K^{-2r} + K \log^2 N \mathbb{E}\left[\|\hat{p}_W - p_W\|_2^2\right].$$

## B.2. Detailed Error Analysis for Theorem 4.2

### B.2.1. ERROR ANALYSIS FOR CASE 1

Recall that $\frac{(2^l mn)^{\frac{1}{2r+2}}}{2^{2S+2}} \leq K < \frac{(2^l mn)^{\frac{1}{2r+2}}}{2^{2S+1}}$, $m \geq n^{2r+1}$ and $1 \leq l \leq \frac{1}{2r+1}\log\frac{m}{n^{2r+1}}$. Then we can verify that $|\mathcal{W}| = 2^{2S+2} \cdot K \geq (2^l mn)^{\frac{1}{2r+2}} > (2^l - 1) \cdot n$ and $m(l \wedge n) \geq m > 4000n\log^2 N$. Hence the condition of 3) in Lemma A.2 is satisfied, and we have $\mathbb{E}[\|\hat{p}_W - p_W\|_2^2] \preceq \frac{2^{2S+2}\cdot K}{2^l mn}$.

Then by (16) in Lemma 4.4 we have

$$\mathbb{E}[\|\bar{f}^H - f\|_2^2] \preceq K^{-2r} + K\log^2 N \frac{2^{2S+2}\cdot K}{mn2^l} \preceq (2^l mn)^{-\frac{r}{r+1}}\log^2 N.$$

### B.2.2. ERROR ANALYSIS FOR CASE 2

Recall that $\frac{(lm)^{\frac{1}{2r+1}}}{2^{2S+2}} \leq K < \frac{(lm)^{\frac{1}{2r+1}}}{2^{2S+1}}$, and it suffices to consider the case for $\frac{2}{2r+1}\log\frac{m}{n^{2r+1}} \vee \frac{n^{2r+1}}{m} \leq l \leq n$. Then we can verify that $(2^l - 1) \cdot n \geq |\mathcal{W}| = 2^{2S+2} \cdot K \geq (lm)^{\frac{1}{2r+1}} \geq n$ and $m(l \wedge n) = ml > 2000n\log^2 N$. Hence the condition of 2) in Lemma A.2 is satisfied, and we have $\mathbb{E}[\|\hat{p}_W - p_W\|_2^2] \preceq \frac{\log(\frac{2^{2S+2}\cdot K}{n}+1)}{ml} \vee \frac{1}{mn}$.

Then by (16) in Lemma 4.4 we have

$$\mathbb{E}[\|\bar{f}^H - f\|_2^2] \preceq K^{-2r} + K\log^2 N \left(\frac{\log(\frac{2^{2S+2}\cdot K}{n}+1)}{ml} \vee \frac{1}{mn}\right) \preceq (lm)^{-\frac{2r}{2r+1}}\log^3 N.$$

### B.2.3. ERROR ANALYSIS FOR CASE 3

First let $m > 2^{2S+2} \cdot 2000\log^2 N$. Recall that $\frac{lm}{2^{2S+2}\cdot 2000\log^2 N} \leq K < \frac{lm}{2^{2S+2}\cdot 1000\log^2 N}$, $n > m^{2r+1}$ and $1 \leq l \leq \frac{n^{\frac{1}{2r+1}}}{m}$. Then we can verify that $|\mathcal{W}| = 2^{2S+2} \cdot K \leq lm \leq n$ and $m(l \wedge |\mathcal{W}|) = ml > 1000|\mathcal{W}|\log^2 N$. Hence the condition of 1) in Lemma A.2 is satisfied, and we have $\mathbb{E}[\|\hat{p}_W - p_W\|_2^2] \preceq \frac{2^{2S+2}\cdot K}{mnl} \vee \frac{1}{mn}$.

By (16) in Lemma 4.4 and $n \geq (ml)^{2r+1}$, then we have

$$\mathbb{E}[\|\bar{f}^H - f\|_2^2] \preceq K^{-2r} + K\log^2 N \left(\frac{2^{2S+2}\cdot K}{mnl} \vee \frac{1}{mn}\right) \preceq (lm)^{-2r}\log^{4r} N.$$

Then let $l > 2^{2S+2} \cdot 2000 \log^2 N$. Recall that $\frac{lm}{2^{2S+2} \cdot 2000 \log^2 N} \leq K < \frac{lm}{2^{2S+2} \cdot 1000 \log^2 N}$, $n > m^{2r+1}$ and $1 \leq l \leq \frac{n^{\frac{1}{2r+1}}}{m}$. In this case we can verify that $l > \log n$, $|\mathcal{W}| = 2^{2S+2} \cdot K \leq lm \leq n$ and $m \lfloor \frac{l}{\lceil \log n \rceil} \rfloor \geq \frac{ml}{2 \log n} \geq |\mathcal{W}|$. Hence the condition of 1') in Lemma A.2 is satisfied, and we have $\mathbb{E}[\|\hat{\boldsymbol{p}}_W - \boldsymbol{p}_W\|_2^2] \preceq \frac{2^{2S+2} \cdot K \log(2^{2S+2} \cdot K)}{mnl} \vee \frac{1}{mn}$.

By (16) in Lemma 4.4 and $n \geq (ml)^{2r+1}$, then we have

$$\mathbb{E}[\|\bar{f}^H - f\|_2^2] \preceq K^{-2r} + K \log^2 N \left( \frac{2^{2S+2} \cdot K \log(2^{2S+2} \cdot K)}{mnl} \vee \frac{1}{mn} \right) \preceq (lm)^{-2r} \log^{4r} N.$$

If $m \leq 2^{2S+2} \cdot 2000 \log^2 N$ and $l \leq 2^{2S+2} \cdot 2000 \log^2 N$, then $ml \preceq \log^4 N$ and the desired bound is vacuous.

### B.2.4. ERROR ANALYSIS FOR CASE 4

Recall that $\frac{(lmn)^{\frac{1}{2r+2}}}{2^{2S+2} \cdot 2000 \log^2 N} \leq K < \frac{(lmn)^{\frac{1}{2r+2}}}{2^{2S+2} \cdot 1000 \log^2 N}$, and it suffices to let $\frac{n^{\frac{1}{2r+1}}}{m} \vee 1 \leq l \leq \frac{n^{2r+1}}{m} \wedge \left( \frac{mn}{2000 \log^2 N} \right)^{\frac{1}{2r+1}}$. Then we can verify that $|\mathcal{W}| = 2^{2S+2} \cdot K \leq (lmn)^{\frac{1}{2r+2}} \leq n$ and $m(l \wedge |\mathcal{W}|) = ml \geq (lmn)^{\frac{1}{2r+2}} > 1000|\mathcal{W}| \log^2 N$. Hence the condition of 1) in Lemma A.2 is satisfied, and we have $\mathbb{E}[\|\hat{\boldsymbol{p}}_W - \boldsymbol{p}_W\|_2^2] \preceq \frac{2^{2S+2} \cdot K}{mnl} \vee \frac{1}{mn}$.

By (16) in Lemma 4.4, then we have

$$\mathbb{E}[\|\bar{f}^H - f\|_2^2] \preceq K^{-2r} + K \log^2 N \left( \frac{2^{2S+2} \cdot K}{mnl} \vee \frac{1}{mn} \right) \preceq (lmn)^{-\frac{r}{r+1}} \log^{4r+2} N.$$

This completes the proof of Theorem 4.2.

## C. Proof of the Lower Bounds in Section 5

### C.1. Proof of Lemma 5.4

Let $\mathcal{P}$ be an $(m, n, l)$-protocol defined in Section 1.3, then we have

$$\sup_{f \in \mathcal{F}} \mathbb{E}[\|\hat{f}_{\mathcal{P}} - f\|_2^2] \geq \sum_{z^k} \frac{1}{2^k} \mathbb{E}[\|\hat{f}_{\mathcal{P}} - f_{z^k}\|_2^2]$$
$$= \mathbb{E}[\|\hat{f}_{\mathcal{P}} - f_{Z^k}\|_2^2].$$

First, we convert the estimation problem into a testing problem by the following procedure. Let

$$\hat{Z}^k = \arg\min_{z^k} \|f_{z^k} - \hat{f}_{\mathcal{P}}\|_2.$$

Then we have

$$\|f_{\hat{Z}^k} - f_{Z^k}\|_2 \leq \|\hat{f}_{\mathcal{P}} - f_{\hat{Z}^k}\|_2 + \|\hat{f}_{\mathcal{P}} - f_{Z^k}\|_2$$
$$\leq 2\|\hat{f}_{\mathcal{P}} - f_{Z^k}\|_2.$$

Hence we have $4\mathbb{E}[\|\hat{f}_{\mathcal{P}} - f_{Z^k}\|_2^2] \geq \mathbb{E}[\|f_{\hat{Z}^k} - f_{Z^k}\|_2^2]$.

Next we establish lower bound for the average testing error. for any $z^k, z'^k \in \{-1, 1\}^k$, since $(\psi_s^k)_{s=1}^k$ have pairwise disjoint supports, we have

$$\|f_{z^k} - f_{z'^k}\|_2^2 = \epsilon^2 k^{-(2r+1)} \sum_{s=1}^k \|\psi_s^k\|_2^2 \cdot \mathbb{1}_{z_s \neq z'_s}$$
$$= \epsilon^2 k^{-(2r+1)} \sum_{s=1}^k \mathbb{1}_{z_s \neq z'_s}.$$

Hence we have

$$\mathbb{E}[\|\hat{f}_{\mathcal{P}} - f_{Z^k}\|_2^2] \geq \frac{1}{4}\mathbb{E}[\|f_{\hat{Z}^k} - f_{Z^k}\|_2^2]$$

$$= \frac{1}{4}\epsilon^2 k^{-(2r+1)} \sum_{s=1}^{k} \mathbb{P}[\hat{Z}_s \neq Z_s]. \tag{33}$$

Since $Z^k - X^{mn} - B^m - \hat{Z}^k$ is a Markov chain, similar to Lemma 10 in (Acharya et al., 2022), we have

$$\frac{1}{k}\sum_{s=1}^{k} I(Z_s; B^m) \geq 1 - h\left(\frac{1}{k}\sum_{s=1}^{k} \mathbb{P}[\hat{Z}_s \neq Z_s]\right), \tag{34}$$

where $h(p) = -p\log_2 p - (1-p)\log_2(1-p)$ is the binary entropy function. By the assumption that $\frac{1}{k}\sum_{s=1}^{k} I(Z_s; B^m) \leq \frac{1}{2}$, then by (34) we have

$$\frac{1}{k}\sum_{s=1}^{k} \mathbb{P}[\hat{Z}_s \neq Z_s] \geq \frac{1}{10}.$$

Thus by (33),

$$\mathbb{E}[\|\hat{f}_{\mathcal{P}} - f_{Z^k}\|_2^2] \succeq k^{-2r}.$$

Then we have $R(m, n, l, r) \succeq k^{-2r}$, completing the proof.

### C.2. Proof of the Centralized Bound for Case 5

Consider the centralized bound $R(m, n, l, r) \succeq (mn)^{-\frac{2r}{2r+1}}$ for Case 5. Note that the bound and the following proof is still valid for all the other cases, though it is not tight except in Case 5. It can be immediately obtained by letting $k = (100mnC_3)^{\frac{1}{2r+1}}$ in the following lemma.

**Lemma C.1.** *Under Assumption 5.1 and 5.2, we have* $\frac{1}{k}\sum_{s=1}^{k} I(Z_s, B^m) \leq \frac{mnC_3}{2k^{2r+1}}$.

Lemma C.1 is derived from Assumption 5.1 and 5.2 as follows.

*Proof.* Let $e_s = \{(-1)^{\delta_{s's}}\}_{s'=1}^{k}$, $s = 1, ..., k$, where $\delta_{s's}$ is equal to 1 if $s' = s$ and 0 otherwise. Let $\odot$ be the element-wise product of sequences, i.e., $\{z_s\}_{s=1}^{k} \odot \{z'_s\}_{s=1}^{k} = \{z_s z'_s\}_{s=1}^{k}$. Then $z^k \odot e_s$ is obtained by only flipping the sign of $z_s$ in $z^k$.

By the Markov chain $Z - X^{mn} - B^m$ and the data processing inequality, we have

$$I(Z_s; B^m) \leq I(Z_s; X^{mn})$$

$$\leq \mathbb{E}\left[D_{\mathcal{X}^{mn}}\left(p_{Z^k}(x^{mn})\|\frac{1}{2}p_{Z^k}(x^{mn}) + \frac{1}{2}p_{Z^k \odot e_s}(x^{mn})\right)\right]$$

$$\leq \frac{1}{2}\mathbb{E}\left[D_{\mathcal{X}^{mn}}\left(p_{Z^k}(x^{mn})\|p_{Z^k \odot e_s}(x^{mn})\right)\right]$$

$$= \frac{1}{2}\mathbb{E}\left[D_{\mathcal{X}^{mn}}\left(p_{Z^k \odot e_s}(x^{mn})\|p_{Z^k}(x^{mn})\right)\right]$$

$$= \frac{mn}{2}\mathbb{E}\left[D_{\mathcal{X}}\left(p_{Z^k \odot e_s}(x)\|p_{Z^k}(x)\right)\right]$$

$$= \frac{mn}{2k}\mathbb{E}\left[D_{[\frac{s-1}{k}, \frac{s}{k}]\times\mathcal{Y}}\left(p_{s,-Z_s}(x)\|p_{s,Z_s}(x)\right)\right]$$

$$= \frac{mn}{2k}\mathbb{E}\left[\mathbb{E}_{p_{s,-Z_s}}\left[-L_{s,-Z_s}(X)|Z_s\right]\right]$$

$$\leq \frac{mnC_3}{2k^{2r+1}},$$

where the second and the third inequality is due to the convexity of KL divergence, the last equality is due to Assumption 5.1 and the last inequality is due to Assumption 5.2. Then we have $\frac{1}{k}\sum_{s=1}^{k} I(Z_s, B^m) \leq \frac{mnC_3}{2k^{2r+1}}$, completing the proof. □

### C.3. Proof for Case 3

We show that $R(m, n, l, r) \preceq (lm)^{-2r}$ for the case 3. Since $Z_s, s = 1, ..., k$ are i.i.d. random variables, we have $I(Z_s; Z_{1:s-1}) = 0$. Then by the chain rule, we have

$$
\begin{aligned}
\sum_{s=1}^{k} I(Z_s; B^m) &\le \sum_{s=1}^{k} I(Z_s; B^m, Z_{1:s-1}) \\
&= \sum_{s=1}^{k} I(Z_s; B^m | Z_{1:s-1}) \\
&= I(Z^k; B^m) \\
&\le H(B^m) \le ml.
\end{aligned}
$$

By letting $k = 100ml$, we have $\frac{1}{k} \sum_{s=1}^{k} I(Z_s; B^m) \le \frac{1}{2}$. This combined with Lemma 5.4 completes the proof.

### C.4. Technical Bounds by the Terminal-Wise Likelihood Ratio

We establish two technical lemmas to prove the remaining bounds. Recall that the terminal-wise likelihood ratio (for testing $z_s$) is defined to be

$$
\mathcal{L}_{s,z^k}(x^n) = \frac{p_{z^k \odot e_s}^n(x^n)}{p_{z^k}^n(x^n)}. \tag{35}
$$

Then by Assumption 5.1, the terminal-wise likelihood ratio $\mathcal{L}_{s,z^k}(x^n)$ can be written as

$$
\mathcal{L}_{s,z^k}(x^n) = \prod_{j=1}^{n} \frac{p_{z^k \odot e_s}(x_j)}{p_{z^k}(x_j)} = \prod_{j:t_j \in [\frac{s-1}{k}, \frac{s}{k}]} \frac{p_{s,-z_s}(x_j)}{p_{s,z_s}(x_j)},
$$

and hence

$$
\log(\mathcal{L}_{s,z^k}(x^n)) = \sum_{j:t_j \in [\frac{s-1}{k}, \frac{s}{k}]} L_{s,z_s}(x_j). \tag{36}
$$

In the following two sections, we recall Lemmas 5.5 and 5.6 and give their proof respectively.

#### C.4.1. EXPONENTIAL BOUND

If $\mathbb{E}_{z^k}[(\mathcal{L}_{s,z^k}(X^n) - 1)^2]$ has an upper bound, then we can get an bound for $\frac{1}{k} \sum_{s=1}^{k} I(Z_s; B^m)$ that is exponential in $l$ as follows.

**Lemma C.2.** *If $\mathbb{E}_{z^k}[(\mathcal{L}_{s,z^k}(X^n) - 1)^2] \le \alpha^2$ for any $s = 1, ..., k$ and $z^k \in \{-1, 1\}^k$, then we have*

$$
\frac{1}{k} \sum_{s=1}^{k} I(Z_s; B^m) \le \frac{2^l m \alpha^2}{2k}. \tag{37}
$$

*Proof.* Similar to the proof of the centralized bound in Appendix C.2, we start by observing that

$$
\begin{aligned}
&I(Z_s; B^m) \\
&\le \mathbb{E}\left[ D_{\{0,1\}^{ml}} \left( p_{Z^k}(b^m) \| \frac{1}{2} p_{Z^k}(b^m) + \frac{1}{2} p_{Z^k \odot e_s}(b^m) \right) \right] \\
&\le \frac{1}{2} \mathbb{E}\left[ D_{\{0,1\}^{ml}} \left( p_{Z^k \odot e_s}(b^m) \| p_{Z^k}(b^m) \right) \right] \\
&= \frac{1}{2} \sum_{i=1}^{m} \mathbb{E}\left[ D_{\{0,1\}^l} \left( p_{Z^k \odot e_s}(b_i | B_{1:i-1}) \| p_{Z^k}(b_i | B_{1:i-1}) \right) \right]
\end{aligned} \tag{38}
$$

Note that

$$p_{Z^k}(b_i|B_{1:i-1}) = \mathbb{E}_{Z^k}[p(b_i|X_i^n, B_{1:i-1})] = \mathbb{E}_{Z^k}[p(b_i|X^n, B_{1:i-1})]$$

and

$$p_{Z^k \odot e_s}(b_i|B_{1:i-1}) = \mathbb{E}_{Z^k \odot e_s}[p(b_i|X^n, B_{1:i-1})] = \mathbb{E}_{Z^k}[\mathcal{L}_{s,z^k}(X^n)p(b_i|X^n, B_{1:i-1})].$$

Then by (25) in Appendix A.2, we have

$$
\begin{aligned}
&D_{\{0,1\}^l}\left(p_{Z^k \odot e_s}(b_i|B_{1:i-1})||p_{Z^k}(b_i|B_{1:i-1})\right) \\
&\leq \chi^2_{\{0,1\}^l}\left(p_{Z^k \odot e_s}(b_i|B_{1:i-1})||p_{Z^k}(b_i|B_{1:i-1})\right) \\
&= \sum_{b_i \in \{0,1\}^l} \frac{\left(\mathbb{E}_{Z^k}[(\mathcal{L}_{s,z^k}(X^n) - 1) \cdot p(b_i|X^n, B_{1:i-1})]\right)^2}{\mathbb{E}_{Z^k}[p(b_i|X^n, B_{1:i-1})]}.
\end{aligned}
\tag{39}
$$

Note that

$$
\begin{aligned}
\mathbb{E}_{z^k}[\mathcal{L}_{s,z^k}(X^n) - 1] &= 0, \\
\mathbb{E}_{z^k}[(\mathcal{L}_{s,z^k}(X^n) - 1)(\mathcal{L}_{s',z^k}(X^n) - 1)] &= 0, \forall s \neq s'.
\end{aligned}
\tag{40}
$$

By (40), $\{1, \mathcal{L}_{1,z^k}(X^n) - 1, \mathcal{L}_{2,z^k}(X^n) - 1, ..., \mathcal{L}_{k,z^k}(X^n) - 1\}$ is an orthogonal system. Then by the Bessel inequality and $\mathbb{E}_{z^k}[(\mathcal{L}_{s,z^k}(X^n) - 1)^2] \leq \alpha^2$ for any $s = 1, ..., k$,

$$
\begin{aligned}
&\sum_{s=1}^{k}\left(\mathbb{E}_{Z^k}[(\mathcal{L}_{s,z^k}(X^n) - 1) \cdot p(b_i|X^n, B_{1:i-1})]\right)^2 \\
&\leq \alpha^2 \mathbb{E}_{Z^k}[p(b_i|X^n, B_{1:i-1})^2] \leq \alpha^2 \mathbb{E}_{Z^k}[p(b_i|X^n, B_{1:i-1})],
\end{aligned}
\tag{41}
$$

where the last inequality is since $p(b_i|X^n, B_{1:i-1}) \in [0, 1]$.

Combining Equations (38), (39) and (41), we have

$$\frac{1}{k}\sum_{s=1}^{k} I(Z_s; B^m) \leq \frac{2^l m \alpha^2}{2k},$$

completing the proof. $\qquad\square$

### C.4.2. POLYNOMIAL BOUND

if $\mathcal{L}_{s,z^k}(X^n)$ is bounded with large probability, then we can get a bound which is a polynomial of $l$, summarized as the following lemma.

**Lemma C.3.** *If there exists some Boolean function $E(x^n)$ such that,*

1. $\mathbb{P}_{z^k}[E(X^n) = 0] \leq \delta_1 < \frac{1}{2}$ *for any* $z^k \in \{-1, 1\}^k$;

2. $E(x^n) = 1$ *if and only if* $|\mathcal{L}_{s,z^k}(x^n) - 1| \leq \delta_2$ *for any* $z^k \in \{-1, 1\}^k$, $s = 1, ..., k$.

*Then we have*

$$\frac{1}{k}\sum_{s=1}^{k} I(Z_s; B^m) \leq m\left((\log 2)\delta_1^{\frac{1}{2}} + \delta_1\right) + \frac{16ml\delta_2^2}{k}.
\tag{42}$$

The nature of Lemma 5.6 is a strong data processing inequality with the strong data processing constant $16\delta_2^2$. To give a tight bound of the constant sufficient for deriving the bound for $\frac{1}{k}\sum_{s=1}^{k} I(Z_s; B^m)$, we need to bound $\delta_1$ and $\delta_2$ simultaneously. The task can be paraphrased as finding a tight bound of $\mathcal{L}_{s,z^k}(X^n) - 1$ that holds with sufficiently large probability. Such a problem is solved by relating it to the balls and bins model, detailed in the next subsection.

*Proof.* By the chain rule of the mutual information, we have

$$I(Z_s; B^m) = \sum_{i=1}^{m} I(Z_s; B_i | B_{1:i-1}). \tag{43}$$

Recall that each sample $x = (t, y) \in [0, 1] \times \mathcal{Y}$. Let $\tilde{t} = \lfloor kt \rfloor$ for such $x$, $\tilde{T} = \lfloor kT \rfloor$ and $\tilde{T}_{ij} = \lfloor kT_{ij} \rfloor$ for $i = 1, ..., m$, $j = 1, ..., n$ correspondingly. By Assumption 5.1, $\tilde{T}_i^n$ is independent of $(Z_s, B_{1:i-1})$, hence $I(Z_s; \tilde{T}_i^n | B_{1:i-1}) = 0$. For each term in the summation in (43), we have

$$I(Z_s; B_i | B_{1:i-1}) \leq I(Z_s; B_i, \tilde{T}_i^n | B_{1:i-1})$$
$$= I(Z_s; \tilde{T}_i^n | B_{1:i-1}) + I(Z_s; B_i | B_{1:i-1}, \tilde{T}_i^n)$$
$$= I(Z_s; B_i | B_{1:i-1}, \tilde{T}_i^n).$$

Then by the assumption 1 on $E(x^n)$,

$$I(Z_s; B_i | B_{1:i-1}, \tilde{T}_i^n) \leq I(Z_s; B_i, E(X_i^n) | B_{1:i-1}, \tilde{T}_i^n)$$
$$= I(Z_s; E(X_i^n) | B_{1:i-1}, \tilde{T}_i^n) + I(Z_s; B_i | B_{1:i-1}, \tilde{T}_i^n, E(X_i^n))$$
$$\leq H(E(X_i^n)) + \mathbb{P}[E(X_i^n) = 0] \cdot I(Z_s; B_i | B_{1:i-1}, \tilde{T}_i^n, E(X_i^n) = 0)$$
$$+ \mathbb{P}[E(X_i^n) = 1] \cdot I(Z_s; B_i | B_{1:i-1}, \tilde{T}_i^n, E(X_i^n) = 1)$$
$$\leq h(\delta_1) + \delta_1 + I(Z_s; B_i | B_{1:i-1}, \tilde{T}_i^n, E(X_i^n) = 1).$$

It suffices to bound the last term above. Before doing that, we first define the conditional terminal-wise likelihood ratio for any $x^n$ with $E(x^n) = 1$ to be

$$\mathcal{L}_{s,z^k}(x^n | \tilde{t}^n, E(x^n) = 1) = \frac{p_{z^k \odot e_s}^n(x^n | \tilde{t}^n, E(x^n) = 1)}{p_{z^k}^n(x^n | \tilde{t}^n, E(x^n) = 1)}, \tag{44}$$

Then we can derive the following useful bound for $\mathcal{L}_{s,z^k}(x^n | \tilde{t}^n, E(x^n) = 1)$. By the assumption 2 on $E(x^n)$,

$$\frac{p_{z^k \odot e_s}^n(\tilde{t}^n, E(x^n) = 1)}{p_{z^k}^n(\tilde{t}^n, E(x^n) = 1)} = \frac{\mathbb{E}_{z^k \odot e_s}\left[\mathcal{L}_{s,z^k}(X^n) \mathbb{1}_{E(X^n)=1, \tilde{T}^n = \tilde{t}^n}\right]}{\mathbb{E}_{z^k}\left[\mathbb{1}_{E(X^n)=1, \tilde{T}^n = \tilde{t}^n}\right]} \in [1 - \delta_2, 1 + \delta_2].$$

Then by the inequality $|ab - 1| \leq |a - 1| + |a| \cdot |b - 1|$,

$$\left| \mathcal{L}_{s,z^k}(x^n | \tilde{t}^n, E(x^n) = 1) - 1 \right|$$
$$= \left| \frac{p_{z^k}^n(\tilde{t}^n, E(x^n) = 1)}{p_{z^k \odot e_s}^n(\tilde{t}^n, E(x^n) = 1)} \cdot \mathcal{L}_{s,z^k}(x^n) - 1 \right|$$
$$\leq \left| \frac{1}{1 - \delta_2} - 1 \right| + \left| \frac{1}{1 - \delta_2} \right| \cdot \delta_2 \tag{45}$$
$$\leq 4\delta_2.$$

Now consider the term $I(Z_s; B_i | B_{1:i-1}, \tilde{T}_i^n, E(X_i^n) = 1)$. Similar to the proof in Appendix C.2,

$$I(Z_s; B_i | B_{1:i-1}, \tilde{T}_i^n, E(X_i^n) = 1)$$
$$\leq \frac{1}{2} \mathbb{E}\left[ D_{\{0,1\}^l}\left( p_{Z^k \odot e_s}(b_i | B_{1:i-1}, \tilde{T}_i^n, E(X_i^n) = 1) \| p_{Z^k}(b_i | B_{1:i-1}, \tilde{T}_i^n, E(X_i^n) = 1) \right) \right]$$
$$= \frac{1}{2} \mathbb{E}\left[ D_{\{0,1\}^l}\left( p_{Z^k \odot e_s}(b_i | B_{1:i-1}, \tilde{T}^n, E(X^n) = 1) \| p_{Z^k}(b_i | B_{1:i-1}, \tilde{T}^n, E(X^n) = 1) \right) \right].$$

In the above equation, further note that

$$p_{Z^k}(b_i|B_{1:i-1}, \tilde{T}^n, E(X^n) = 1) = \mathbb{E}_{p_{Z^k}(\cdot|\tilde{T}^n, E(X^n)=1)}[p(b_i|X^n, B_{1:i-1})]$$

and

$$p_{Z^k \odot e_s}(b_i|B_{1:i-1}, \tilde{T}^n, E(X^n) = 1)$$
$$= \mathbb{E}_{p_{Z^k}(\cdot|\tilde{T}^n, E(X^n)=1)}[\mathcal{L}_{s,z^k}(X^n|\tilde{T}^n, E(X^n) = 1)p(b_i|X^n, B_{1:i-1})].$$

Then by (25) in Appendix A.2, we have

$$D_{\{0,1\}^l}\left(p_{Z^k \odot e_s}(b_i|B_{1:i-1}, \tilde{T}^n, E(X^n) = 1)||p_{Z^k}(b_i|B_{1:i-1}, \tilde{T}^n, E(X^n) = 1)\right)$$
$$\leq \chi^2_{\{0,1\}^l}\left(p_{Z^k \odot e_s}(b_i|B_{1:i-1}, \tilde{T}^n, E(X^n) = 1)||p_{Z^k}(b_i|B_{1:i-1}, \tilde{T}^n, E(X^n) = 1)\right)$$
$$= \sum_{b_i \in \{0,1\}^l} \frac{\left(\mathbb{E}_{p_{Z^k}(\cdot|\tilde{T}^n, E(X^n)=1)}[(\mathcal{L}_{s,z^k}(X^n|\tilde{T}^n, E(X^n) = 1) - 1)p(b_i|X^n, B_{1:i-1})]\right)^2}{\mathbb{E}_{p_{Z^k}(\cdot|\tilde{T}^n, E(X^n)=1)}[p(b_i|X^n, B_{1:i-1})]}$$

Combining these results, we can obtain that

$$\frac{1}{k}\sum_{s=1}^{k} I(Z_s; B^m) \leq m(h(\delta_1) + \delta_1)$$
$$+ \frac{1}{2k}\sum_{i=1}^{m} \mathbb{E}\left[\sum_{b_i \in \{0,1\}^l}\sum_{s=1}^{k} \frac{\left(\mathbb{E}_{p_{Z^k}(\cdot|\tilde{T}^n, E(X^n)=1)}[(\mathcal{L}_{s,z^k}(X^n|\tilde{T}^n, E(X^n) = 1) - 1)p(b_i|X^n, B_{1:i-1})]\right)^2}{\mathbb{E}_{p_{Z^k}(\cdot|\tilde{T}^n, E(X^n)=1)}[p(b_i|X^n, B_{1:i-1})]}\right]. \tag{46}$$

The following transportation lemma is useful for bounding the last term. It can be proved by well-known arguments (cf. Remark 4.25 in (van Handel, 2016) and Lemma 3 in (Acharya et al., 2023)). The definition and related properties of sub-Gaussian random variables can be found in Appendix A.1.

**Lemma C.4** (Transportation Lemma). *Let $P$ and $Q$ be two measures on a probability space $\mathcal{U}$ and $P \ll Q$. $U$ is a sub-Gaussian random vector with parameter $\sigma$ under the measure $P$. Then we have*

$$\|\mathbb{E}_Q[U]\|_2^2 \leq 2\sigma^2 D(Q||P). \tag{47}$$

Now let $X^n$ be i.i.d. random variables and each $X_i$ is generated following the distribution $p_{Z^k}(\cdot|\tilde{T}^n, E(X^n) = 1)$. Then by Assumption 5.1 and (23), $\mathcal{L}_{s,z^k}(X^n|\tilde{T}^n, E(X^n) = 1)$ for $s = 1, ..., k$ are independent. By the discussion in Appendix A.1, since each $\mathcal{L}_{s,z^k}(X^n|\tilde{T}^n, E(X^n) = 1) - 1$ is bounded by $\sigma = 4\delta_2$ in (45) and has zero mean, it is sub-Gaussian with parameter $\sigma$. Then we have $(\mathcal{L}_{s,z^k}(X^n|\tilde{T}^n, E(X^n) = 1) - 1)_{s=1}^{k}$ is a sub-Gaussian random vector with parameter $\sigma$. Applying the transportation lemma to $U = (\mathcal{L}_{s,z^k}(X^n|\tilde{T}^n, E(X^n) = 1) - 1)_{s=1}^{k}$, $P = p_{Z^k}(\cdot|\tilde{T}^n, E(X^n) = 1)$ and $\frac{dQ}{dP} = \frac{p(b_i|\cdot, B_{1:i-1})}{\mathbb{E}_P[p(b_i|\cdot, B_{1:i-1})]}$, then we have

$$\sum_{s=1}^{k} \frac{\left(\mathbb{E}_{p_{Z^k}(\cdot|\tilde{T}^n, E(X^n)=1)}\left[(\mathcal{L}_{s,z^k}(X^n|\tilde{T}^n, E(X^n) = 1) - 1)p(b_i|X^n, B_{1:i-1})\right]\right)^2}{\mathbb{E}_{p_{Z^k}(\cdot|\tilde{T}^n, E(X^n)=1)}[p(b_i|X^n, B_{1:i-1})]}$$
$$\leq 16\delta_2^2 \cdot \mathbb{E}_{p_{Z^k}(\cdot|\tilde{T}^n, E(X^n)=1)}\left[p(b_i|X^n, B_{1:i-1}) \log\left(\frac{p(b_i|X^n, B_{1:i-1})}{\mathbb{E}_{p_{Z^k}(\cdot|\tilde{T}^n, E(X^n)=1)}[p(b_i|X^n, B_{1:i-1})]}\right)\right].$$

This combined with (46) imply that

$$
\frac{1}{k} \sum_{s=1}^{k} I(Z_s; B^m)
$$

$$
\leq m(h(\delta_1) + \delta_1) + \frac{16\delta_2^2}{k} \cdot \sum_{i=1}^{m} I(B_i; X_i^n | B_{1:i-1}, \tilde{T}_i^n, E(X^n) = 1)
$$

$$
\leq m((\log 2)\delta_1^{\frac{1}{2}} + \delta_1) + \frac{16\delta_2^2}{k} \cdot \sum_{i=1}^{m} H(B_i)
$$

$$
\leq m((\log 2)\delta_1^{\frac{1}{2}} + \delta_1) + \frac{16ml\delta_2^2}{k},
$$

completing the proof. $\qquad\square$

### C.5. Detailed Proof for Case 1

By Assumption 5.2, since $\frac{1}{\beta} = O(k^r) \gg 2$ we have

$$
\mathbb{E}_{p_{s,z_s}}[\exp(2L_{s,z_s}(X))] \leq \exp(2\nu^2).
$$

Then by (23) and $1 + x \leq e^x \leq 1 + 2x$ for $0 \leq x \leq 1$,

$$
\begin{aligned}
\mathbb{E}_{z^k}[(\mathcal{L}_{s,z^k}(X^n))^2] &= \mathbb{E}_{z^k}\left[\exp\left(\sum_{j:T_j \in [\frac{s-1}{k}, \frac{s}{k}]} 2L_{s,z_s}(X_j)\right)\right] \\
&= \mathbb{E}\left[\left(\mathbb{E}_{p_{s,z_s}}[\exp\left(2L_{s,z_s}(X)\right)]\right)^{\sum_{j=1}^{n} \mathbb{1}_{T_j \in [\frac{s-1}{k}, \frac{s}{k}]}}\right] \\
&\leq \mathbb{E}\left[\exp\left(2\nu^2 \cdot \sum_{j=1}^{n} \mathbb{1}_{T_j \in [\frac{s-1}{k}, \frac{s}{k}]}\right)\right] \\
&= \sum_{r=0}^{n} \binom{n}{r} \left(\frac{1}{k}\right)^r \left(1 - \frac{1}{k}\right)^{n-r} \exp(2\nu^2 r) \\
&= \left[1 + \frac{1}{k}\left(\exp(2\nu^2) - 1\right)\right]^n \\
&\leq \exp\left(\frac{n}{k}\left(\exp(2\nu^2) - 1\right)\right) \\
&\leq \exp\left(\frac{4n\nu^2}{k}\right).
\end{aligned}
$$

Then by (40), we have

$$
\mathbb{E}_{z^k}[(\mathcal{L}_{s,z^k}(X^n) - 1)^2] = \mathbb{E}_{z^k}[(\mathcal{L}_{s,z^k}(X^n))^2] - 1 \leq \exp\left(\frac{4n\nu^2}{k}\right) - 1 \leq \frac{8n\nu^2}{k},
$$

as long as $\frac{4n\nu^2}{k} = \frac{4C_1 n}{k^{2r+1}} < 1$. Let $k = (C_4 mn2^l)^{\frac{1}{2r+2}}$ and $\alpha^2 = \frac{8n\nu^2}{k} = \frac{8C_1 n}{k^{2r+1}}$, where $C_4$ is a big constant. Then by $m > n^{2r+1}$, we have $k > (C_4 mn)^{\frac{1}{2r+2}} > (C_4)^{\frac{1}{2r+2}} n$ and hence $\frac{4n\nu^2}{k} < 1$ is satisfied. By Lemma 5.5, we have

$$
\frac{1}{k} \sum_{s=1}^{k} I(Z_s; B^m) \leq \frac{2^{l+2} mn}{k^{2r+2}} < \frac{1}{2}.
$$

Then by Lemma 5.4, we obtain that $R(m, n, l, r) \succeq (mn2^l)^{-\frac{r}{r+1}}$.

## C.6. Detailed Proof for Case 2

Recall that

$$1 \vee \frac{n^{2r+1}}{m} \leq l \leq n. \tag{48}$$

Let $k = (1000C_5^2 C_2^2 ml \log^2 m)^{\frac{1}{2r+1}}$, where $C_5$ is a large constant to be determined, and then we have

$$n \vee m^{\frac{1}{2r+1}} \leq k \leq m^{\frac{1}{r}}. \tag{49}$$

Let

$$E(x^n) = \begin{cases} 1, & \text{if } \left|\mathcal{L}_{s,z^k}(x^n) - 1\right| \leq 2C_5\beta \log m, \forall z^k \in \{-1,1\}^k, s = 1, ..., k, \\ 0, & \text{otherwise.} \end{cases}$$

Then we have $\delta_2 \leq 2C_5\beta \log m$ immediately. To use Lemma 5.6, we need to bound the quantity $\mathbb{P}_{z^k}[E(X^n) = 0]$ for any $z^k$. Note that by (23) and $2C_5\beta \log m \asymp k^{-r} \log m \to 0$, we have

$$\{E(X^n) = 0\} = \left\{\left|\mathcal{L}_{s,z^k}(X^n) - 1\right| \leq 2C_5\beta \log m, \forall z^k \in \{-1,1\}^k, s = 1, ..., k\right\}^{\complement}$$

$$\subseteq \left\{\left|\log \mathcal{L}_{s,z^k}(X^n)\right| < C_5\beta \log m, \forall z^k \in \{-1,1\}^k, s = 1, ..., k\right\}^{\complement}$$

$$= \left\{\left|\sum_{j:T_j \in [\frac{s-1}{k}, \frac{s}{k}]} L_{s,z_s}(X_j)\right| < C_5\beta \log m, \forall s = 1, ..., k, z_s \in \{-1,1\}\right\}^{\complement}$$

$$= \left(\bigcap_{s=1}^{k} \bigcap_{z_s=\pm 1} \left\{\left|\sum_{j:T_j \in [\frac{s-1}{k}, \frac{s}{k}]} L_{s,z_s}(X_j)\right| < C_5\beta \log m\right\}\right)^{\complement} \tag{50}$$

$$= \bigcup_{s=1}^{k} \bigcup_{z_s=\pm 1} \left\{\left|\sum_{j:T_j \in [\frac{s-1}{k}, \frac{s}{k}]} L_{s,z_s}(X_j)\right| \geq C_5\beta \log m\right\}.$$

Define the event $\mathcal{E}_{s,z_s} = \left\{\left|\sum_{j:T_j \in [\frac{s-1}{k}, \frac{s}{k}]} L_{s,z_s}(X_j)\right| \geq C_5\beta \log m\right\}$. For any $x^n$ where $x_j = (t_j, y_j)$, define the empirical distribution for $(\lfloor kt_j \rfloor)_{j=1}^n$ to be

$$V_s(x^n) = \sum_{j=1}^{n} \mathbb{1}_{t_j \in [\frac{s-1}{k}, \frac{s}{k}]}, \tag{51}$$

for any $s = 1, ..., k$. Then define $E'(x^n)$ based on $(V_s(x^n))_{s=1}^k$ by

$$E'(x^n) = \begin{cases} 1, & \text{if } V_s(x^n) \leq 200 \log m, \forall s = 1, ..., k, \\ 0, & \text{otherwise.} \end{cases}$$

Note that by the union bound and (50),

$$\mathbb{P}_{z^k}[E(X^n) = 0] \leq \mathbb{P}_{z^k}[E'(X^n) = 0] + \sum_{s=1}^{k} \sum_{z_s=\pm 1} \mathbb{P}_{z^k}[\mathcal{E}_{s,z_s} | E'(X^n) = 1]. \tag{52}$$

The first term can be bounded by letting $c = C_6 \log m$ in (26) where $C_6$ is a large constant to be determined, which yields

$$\mathbb{E}_{z^k}[E'(X^n) = 0] \leq k \exp\left(-\frac{C_6}{2} \log m\right) = km^{-\frac{C_6}{2}}. \tag{53}$$

Now consider each term in the latter summation. Conditioned on $\lfloor kT_j \rfloor, j = 1, ..., n$, $(X_j)_{j=1}^n$ are independent. Furthermore, on the event $\{E'(X^n) = 1\}$, $V_s(X^n) \leq C_6 \log m$ for any $s = 1, ..., k$. So we have

$$\mathbb{P}_{z^k}\left[\mathcal{E}_{s,z_s} \Big| \lfloor kT_j \rfloor, j = 1, ..., n, E'(X^n) = 1\right] = \mathbb{P}\left[\left|\sum_{j=1}^{V_s(X^n)} L_{s,z_s}(X'_j)\right| \geq C_5\beta \log m\right],$$

where $(X'_j)_{j=1}^n$ are i.i.d. random variables generated from the distribution $p_{s,z_s}$. By Assumption 5.2, $\sum_{j=1}^{V_s(X^n)} L_{s,z_s}(X'_j)$ is sub-exponential with parameters $(\sqrt{C_6\nu^2\log m}, \beta)$ and $\mathbb{E}\left[\left|\sum_{j=1}^{V_s(X^n)} L_{s,z_s}(X'_j)\right|\right] \leq \frac{C_6 C_3 \log m}{k^{2r}}$. By (49), we have

$$\frac{C_6\nu^2\log m}{\beta} = \frac{C_6 C_1 k^{-2r}\log m}{C_2 k^{-r}} \leq C_5 C_2 k^{-r}\log m = C_5\beta\log m$$

as long as $C_5 > \frac{C_6 C_1}{C_2^2}$, and

$$C_5\beta\log m \asymp k^{-r}\log m \gg k^{-2r}\log m \asymp \frac{C_6 C_3 \log m}{k^{2r}}.$$

Then Hoeffding's inequality (Lemma A.1 in Appendix A.1) implies that

$$\mathbb{P}_{z^k}\left[\mathcal{E}_{s,z_s}\middle| \lfloor kT_j\rfloor, j=1,...,n, E'(X^n)=1\right] \leq 2\exp\left(-\frac{\frac{1}{2}C_5\beta\log m}{2\beta}\right) = 2m^{-\frac{C_5}{4}}.$$

Thus we have

$$\begin{aligned}
&\mathbb{P}_{z^k}[\mathcal{E}_{s,z_s}|E'(X^n)=1]\\
=&\mathbb{E}_{z^k}\left[\mathbb{P}_{z^k}\left[\mathcal{E}_{s,z_s}\middle|\lfloor kT_j\rfloor, j=1,...,n, E'(X^n)=1\right]\right]\\
\leq& 2m^{-\frac{C_5}{4}}.
\end{aligned} \tag{54}$$

Combining the bound (52), (53) and (54), we have

$$\mathbb{P}_{z^k}[E(X^n)=0] \leq km^{-\frac{C_6}{2}} + 4km^{-\frac{C_5}{4}} \leq m^{-100}$$

where the last inequality is by (49), as long as $C_5$ and $C_6$ are both big enough. Then we can choose $\delta_1 = m^{-100}$.

By Lemma 5.6, the inequality (48) and (49), we finally have

$$\begin{aligned}
\frac{1}{k}\sum_{s=1}^k I(Z_s; B^m) &\leq m((\log 2)\delta_1^{\frac{1}{2}} + \delta_1) + \frac{16ml\delta_2^2}{k}\\
&\leq 4m^{-50+1} + \frac{64C_5^2 ml\beta^2\log^2 m}{k}\\
&\leq 4m^{-49} + \frac{64C_5^2 C_2^2 ml\log^2 m}{k^{2r+1}}\\
&\leq \frac{1}{2},
\end{aligned}$$

where the last inequality is by the choice of $k = (1000C_5^2 C_2^2 ml\log^2 m)^{\frac{1}{2r+1}}$. Then by Lemma 5.4, we complete the proof.

## C.7. Detailed Proof for Case 4

Recall that $r > \frac{1}{2}$ and

$$\frac{n^{\frac{1}{2r+1}}}{m}\vee 1 \leq l \leq \frac{n^{2r+1}}{m}\wedge(mn)^{\frac{1}{2r+1}}\leq n. \tag{55}$$

Let $k = (1000C_1 C_7 mnl)^{\frac{1}{2r+2}}\log^{\frac{2r+3}{2r+1}} n$, where $C_7$ is a large constant to be determined, and then we have

$$k \ll n\log^2 n \leq n^2 \tag{56}$$

and

$$k^{2r+1} \succeq (mnl)^{\frac{2r+1}{2r+2}}\log^{2r+3} n \succeq n\log^{2r+3} n \gg n\log^4 n. \tag{57}$$

Let

$$E(x^n) = \begin{cases} 1, \text{ if } |\mathcal{L}_{s,z^k}(x^n) - 1| \leq 2\sqrt{\dfrac{C_7\nu^2 n\log^4 n}{k}}, \forall z^k \in \{-1,1\}^k, s=1,...,k,\\ 0, \text{ otherwise.}\end{cases}$$

Then we have $\delta_2 \leq 2\sqrt{\frac{C_7\nu^2 n \log^4 n}{k}}$ immediately. To use Lemma 5.6, we first bound the quantity $\mathbb{P}_{z^k}[E(X^n) = 0]$ for any $z^k$. Note that by (57), $\sqrt{\frac{C_7\nu^2 n \log^4 n}{k}} \asymp \sqrt{\frac{n \log^4 n}{k^{2r+1}}} \to 0$. Then by (23) and similar to (50) we have

$$\{E(X^n) = 0\} \subseteq \bigcup_{s=1}^{k} \bigcup_{z_s=\pm 1} \left\{ \left| \sum_{j:T_j \in [\frac{s-1}{k}, \frac{s}{k}]} L_{s,z_s}(X_j) \right| \geq \sqrt{\frac{C_7\nu^2 n \log^4 n}{k}} \right\}. \tag{58}$$

Let $\mathcal{E}_{s,z_s} = \left\{ \left| \sum_{j:T_j \in [\frac{s-1}{k}, \frac{s}{k}]} L_{s,z_s}(X_j) \right| \geq \sqrt{\frac{C_7\nu^2 n \log^4 n}{k}} \right\}$. Define $E'(x^n)$ based on $(V_s(x^n))_{s=1}^{k}$ (cf. (51)) by

$$E'(x^n) = \begin{cases} 1, & \text{if } V_s(x^n) < \dfrac{C_8 n \log^3 n}{k}, \forall s = 1, ..., k, \\ 0, & \text{otherwise.} \end{cases}$$

Then by Equation (27), we have

$$\mathbb{E}_{z^k}[E'(X^n) = 0] \leq k \exp\left(-\frac{C_8 n \log^3 n}{8k}\right) \leq k \exp\left(-\frac{C_8 \log n}{8}\right) = kn^{-\frac{C_8}{8}}. \tag{59}$$

$(X_j)_{j=1}^{n}$ are independent given $\lfloor kT_j \rfloor$, $j = 1, ..., n$. And on the event $\{E'(X^n) = 1\}$, $V_s(X^n) \leq \frac{C_8 n \log^3 n}{k}$ for any $s = 1, ..., k$. So we have

$$\mathbb{P}_{z^k}\left[\mathcal{E}_{s,z_s} \Big| \lfloor kT_j \rfloor, j = 1, ..., n, E'(X^n) = 1\right] = \mathbb{P}\left[\left| \sum_{j=1}^{V_s(X^n)} L_{s,z_s}(X'_j) \right| \geq \sqrt{\frac{C_7\nu^2 n \log^4 n}{k}}\right],$$

where $(X'_j)_{j=1}^{n}$ are i.i.d. copies of $X^n$. By Assumption 5.2, $\sum_{j=1}^{V_s(X^n)} L_{s,z_s}(X'_j)$ is sub-exponential with parameters $\left(\sqrt{\frac{C_8\nu^2 n \log^3 n}{k}}, \beta\right)$ and $\mathbb{E}\left[\left| \sum_{j=1}^{V_s(X^n)} L_{s,z_s}(X'_j) \right|\right] \leq \frac{C_3 C_8 n \log^3 n}{k^{2r+1}}$. By (56), we have

$$\frac{C_8\nu^2 n \log^3 n}{k\beta} \asymp \frac{n \log^3 n}{k^{r+1}} \gg \sqrt{\frac{n \log^4 n}{k^{2r+1}}} \asymp \sqrt{\frac{C_7\nu^2 n \log^4 n}{k}}$$

and by (57), we have

$$\sqrt{\frac{C_7\nu^2 n \log^4 n}{k}} \asymp \sqrt{\frac{n \log^4 n}{k^{2r+1}}} \succeq \frac{n \log^3 n}{k^{2r+1}} \asymp \frac{C_3 C_8 n \log^3 n}{k^{2r+1}}.$$

Then Lemma A.1 in Appendix A.1 implies that

$$\mathbb{P}_{z^k}\left[\mathcal{E}_{s,z_s} \Big| \lfloor kT_j \rfloor, j = 1, ..., n, E'(X^n) = 1\right] \leq 2\exp\left(-\frac{\frac{C_7\nu^2 n \log^4 n}{4k}}{\frac{2C_8\nu^2 n \log^3 n}{k}}\right) = 2\exp\left(-\frac{C_7 \log n}{8C_8}\right) = 2n^{-\frac{C_7}{8C_8}}.$$

Thus we have

$$\mathbb{P}_{z^k}[\mathcal{E}_{s,z_s} | E'(X^n) = 1] \leq 2n^{-\frac{C_7}{8C_8}}. \tag{60}$$

Combining the bounds (58), (59) and (60), we have

$$\mathbb{P}_{z^k}[E(X^n) = 0] \leq \mathbb{P}_{z^k}[E'(X^n) = 0] + \sum_{s=1}^{k} \sum_{z_s=\pm 1} \mathbb{P}_{z^k}[\mathcal{E}_{s,z_s} | E'(X^n) = 1]$$

$$\leq kn^{-\frac{C_8}{8}} + 4kn^{-\frac{C_7}{8C_8}} \leq n^{-100(r+1)},$$

where the last inequality is by (56), as long as $\frac{C_7}{C_8}$ and $C_8$ are both big enough. Then we can choose $\delta_1 = n^{-100(r+1)}$.

By Lemma 5.6, (55) and (57), we finally have

$$
\begin{aligned}
\frac{1}{k} \sum_{s=1}^{k} I(Z_s; B^m) &\leq n^{2r+1}((\log 2)\delta_1^{\frac{1}{2}} + \delta_1) + \frac{16ml\delta_2^2}{k} \\
&\leq 4n^{-50+1} + \frac{64C_7 mnl\nu^2 \log^4 n}{k^2} \\
&\leq 4n^{-49} + \frac{64C_1 C_7 mnl \log^4 n}{k^{2r+2}} \\
&\leq \frac{1}{2},
\end{aligned}
$$

where the last inequality is by the choice of $k = (1000C_1 C_7 mnl)^{\frac{1}{2r+2}} \log^{\frac{2r+3}{2r+1}} n$ and $r > \frac{1}{2}$. By Lemma 5.4, we complete the proof.

## D. Proof for the Specific Cases in Section 2.3

### D.1. Proof of Corollary 2.7

Let $T = X$ and $Y = 1$, then $X = (T, Y)$.

#### D.1.1. VERIFICATION OF ASSUMPTION 4.1

Let $h(y) = y$, then $\hat{f}_{Hs}(X) = \phi_{Hs}(X)$. By 1) in Lemma 3.1, we have $|\hat{f}_{Hs}(X)| \leq C \cdot 2^{\frac{H}{2}} = C\sqrt{K}$ for some $C > 0$, hence $\hat{f}_{Hs}(X)$ is sub-exponential with parameters $(\sqrt{C^2 K}, 0)$. Moreover, we have

$$
\mathbb{E}[\hat{f}_{Hs}(X)] = \int_0^1 f(x)\phi_{Hs}(x)dx = f_{Hs},
$$

hence $\hat{f}_{Hs}(X)$ is an unbiased estimator of $f_{Hs}$.

#### D.1.2. VERIFICATION OF ASSUMPTION 5.1

For any $s = 1, ..., k$, $z_s = \pm 1$ and $x \in [\frac{s-1}{k}, \frac{s}{k}]$, we have

$$
p_{z^k}(x) = f_{z_k}(x) = 1 + \epsilon k^{-(r+\frac{1}{2})} \sum_{s'=1}^{k} z_s \psi_{s'}^k(x) = \frac{1}{k} \cdot k \left(1 + \epsilon k^{-(r+\frac{1}{2})} z_s \psi_s^k(x)\right).
$$

Note that

$$
\int_{\frac{s-1}{k}}^{\frac{s}{k}} k \left(1 + \epsilon k^{-(r+\frac{1}{2})} z_s \psi_s^k(x)\right) dx = 1,
$$

and by 1) in Lemma 3.1, $1 + \epsilon k^{-(r+\frac{1}{2})} z_s \psi_s^k(x) \geq 1 - \|\epsilon k^{-(r+\frac{1}{2})} z_s \psi_s^k\|_\infty \succeq 1 - k^{-r} > 0$. Hence $p_{s,z_s}(x) = k\left(1 + \epsilon k^{-(r+\frac{1}{2})} z_s \psi_s^k(x)\right)$ is a distribution function on $[\frac{s-1}{k}, \frac{s}{k}]$.

#### D.1.3. VERIFICATION OF ASSUMPTION 5.2

For any $s = 1, ..., k$, $z_s = \pm 1$ and $x \in [\frac{s-1}{k}, \frac{s}{k}]$,

$$
L_{s,z_s}(x) = \log \left(\frac{1 - \epsilon k^{-(r+\frac{1}{2})} z_s \psi_s^k(x)}{1 + \epsilon k^{-(r+\frac{1}{2})} z_s \psi_s^k(x)}\right) = \log \left(1 - \frac{2\epsilon k^{-(r+\frac{1}{2})} z_s \psi_s^k(x)}{1 + \epsilon k^{-(r+\frac{1}{2})} z_s \psi_s^k(x)}\right).
$$

Then for $X \sim p_{s,z_s}$, by 1) in Lemma 3.1, $|L_{s,z_s}(X)| \leq C'k^{-r}$ for some $C' > 0$, hence $L_{s,z_s}(X)$ is sub-exponential with parameters $(\sqrt{C^2 k^{-2r}}, 0)$. Moreover, by the inequality $-x - x^2 \leq \log(1-x) \leq -x$ for $|x| < \frac{1}{2}$, then

$$
\left| \mathbb{E}[L_{s,z_s}(X)] \right| \leq \left| \mathbb{E}\left[ \frac{2\epsilon k^{-(r+\frac{1}{2})} z_s \psi_s^k(X)}{1 + \epsilon k^{-(r+\frac{1}{2})} z_s \psi_s^k(X)} \right] \right| + \mathbb{E}\left[ \left( \frac{2\epsilon k^{-(r+\frac{1}{2})} z_s \psi_s^k(X)}{1 + \epsilon k^{-(r+\frac{1}{2})} z_s \psi_s^k(X)} \right)^2 \right]
$$

$$
\preceq 0 + k^{-(2r+1)} \cdot 2^h \asymp k^{-2r},
$$

completing the proof.

### D.2. Proof of Corollary 2.12

In all these cases, $X = (T, Y)$ with $T \in [0, 1]$.

### 1. Nonparametric Gaussian Regression Problem in Example 2.8

#### D.2.1. VERIFICATION OF ASSUMPTION 4.1

Let $h(y) = y$, then $\hat{f}_{Hs}(X) = \phi_{Hs}(T)Y$. Note that

$$
\mathbb{E}[\hat{f}_{Hs}(X)] = \mathbb{E}[\phi_{Hs}(T)\mathbb{E}[Y|T]] = \mathbb{E}[\phi_{Hs}(T)f(T)] = \int_0^1 f(t)\phi_{Hs}(t)dt = f_{Hs},
$$

hence $\hat{f}_{Hs}(X)$ is an unbiased estimator of $f_{Hs}$.

By 1) in Lemma 3.1, we have $|\phi_{Hs}(T)| \leq C \cdot 2^{\frac{H}{2}} = C\sqrt{K}$ for some $C > 0$. Then by 4) in Lemma 3.1, $|\phi_{Hs}(T)f(T)| \leq CL'\sqrt{K}$ almost surely, hence $\phi_{Hs}(T)f(T)$ is sub-Gaussian with parameter $CL'\sqrt{K}$. Since $Y|T \sim \mathcal{N}(f(T), 1)$, we have $\mathbb{E}\left[ e^{\lambda \phi_{Hs}(T)(Y-f(T))} \big| T \right] = e^{\frac{1}{2}(\lambda \phi_{Hs}(T))^2}, \forall \lambda \in \mathbb{R}$. Then for any $\lambda \in \mathbb{R}$, we have

$$
\begin{aligned}
\mathbb{E}\left[ e^{\lambda(\hat{f}_{Hs}(X)-f_{Hs})} \right] &= \mathbb{E}\left[ \mathbb{E}\left[ e^{\lambda \phi_{Hs}(T)(Y-f(T))} \big| T \right] e^{\lambda(\phi_{Hs}(T)f(T)-f_{Hs})} \right] \\
&= \mathbb{E}\left[ e^{\frac{1}{2}(\lambda \phi_{Hs}(T))^2} e^{\lambda(\phi_{Hs}(T)f(T)-f_{Hs})} \right] \\
&\leq e^{\frac{1}{2}C^2 K \lambda^2} \mathbb{E}\left[ e^{\lambda(\phi_{Hs}(T)f(T)-f_{Hs})} \right] \\
&\leq e^{\frac{1}{2}C^2 K(1+L'^2)\lambda^2},
\end{aligned}
$$

implying that $\hat{f}_{Hs}(X)$ is sub-exponential with parameters $(\sqrt{C^2 K(1 + L'^2)}, 0)$.

#### D.2.2. VERIFICATION OF ASSUMPTION 5.1

For any $s = 1, ..., k$, $z_s = \pm 1$ and $x = (t, y)$ with $t \in [\frac{s-1}{k}, \frac{s}{k}]$, we have

$$
p_{z^k}(x) = \frac{1}{\sqrt{2\pi}} e^{-\frac{(y-f_{z_k}(t))^2}{2}} == \frac{1}{k} \cdot \frac{k}{\sqrt{2\pi}} e^{-\frac{(y-\epsilon k^{-(r+\frac{1}{2})} z_s \psi_s^k(t))^2}{2}}.
$$

Note that

$$
\int_{\frac{s-1}{k}}^{\frac{s}{k}} \int_{\mathbb{R}} \frac{k}{\sqrt{2\pi}} e^{-\frac{(y-\epsilon k^{-(r+\frac{1}{2})} z_s \psi_s^k(t))^2}{2}} dy\, dt = 1,
$$

Hence $p_{s,z_s}(x) = \frac{k}{\sqrt{2\pi}} e^{-\frac{(y-\epsilon k^{-(r+\frac{1}{2})} z_s \psi_s^k(t))^2}{2}} > 0$ is a distribution function on $[\frac{s-1}{k}, \frac{s}{k}] \times \mathbb{R}$.

#### D.2.3. VERIFICATION OF ASSUMPTION 5.2

For any $s = 1, ..., k$, $z_s = \pm 1$ and $x = (t, y)$ with $t \in [\frac{s-1}{k}, \frac{s}{k}]$,

$$
L_{s,z_s}(x) = \left( y + \epsilon k^{-(r+\frac{1}{2})} z_s \psi_s^k(t) \right)^2 - \left( y - \epsilon k^{-(r+\frac{1}{2})} z_s \psi_s^k(t) \right)^2 = 4\epsilon k^{-(r+\frac{1}{2})} z_s y \psi_s^k(t).
$$

Then for $X = (T, Y) \sim p_{s,z_s}$, $L_{s,z_s}(X) = 4\epsilon k^{-(r+\frac{1}{2})} z_s Y \psi_s^k(T)$. Similar to the verification of Assumption 4.1, $Y \psi_s^k(T)$ is sub-exponential with parameters $(\sqrt{C^2 k(1 + L'^2)}, 0)$ for some $C > 0$. Thus $L_{s,z_s}(X)$ is sub-exponential with parameters $(\sqrt{C^2 k^{-2r}(1 + L'^2)}, 0)$.

Moreover, by 3) in Lemma 3.1 we have

$$
\begin{aligned}
|\mathbb{E}[L_{s,z_s}(X)]| &= 4\epsilon k^{-(r+\frac{1}{2})} \left| \mathbb{E}\left[ \mathbb{E}[Y|T] \psi_s^k(T) \right] \right| \\
&= 4\epsilon k^{-(r+\frac{1}{2})} \left| \mathbb{E}\left[ f(T) \psi_s^k(T) \right] \right| \\
&= 4\epsilon k^{-(r+\frac{1}{2})} f_{h-h_0,s} \\
&\leq 4\epsilon k^{-(r+\frac{1}{2})} \| f - f^{h-h_0-1} \|_2 \\
&\preceq k^{-(r+\frac{1}{2})} \sqrt{k^{-2r}} \preceq k^{-2r},
\end{aligned}
$$

completing the proof.

## 2. Nonparametric Binary Regression Problem in Example 2.9

### D.2.1. VERIFICATION OF ASSUMPTION 4.1

Let $h(y) = y$, then $\hat{f}_{Hs}(X) = \phi_{Hs}(T)Y$. Note that

$$
\mathbb{E}[\hat{f}_{Hs}(X)] = \mathbb{E}[\phi_{Hs}(T)\mathbb{E}[Y|T]] = \mathbb{E}[\phi_{Hs}(T)f(T)] = \int_0^1 f(t)\phi_{Hs}(t)dt = f_{Hs},
$$

hence $\hat{f}_{Hs}(X)$ is an unbiased estimator of $f_{Hs}$. Moreover, by 1) in Lemma 3.1 and $Y \in \{0, 1\}$, we have $|\hat{f}_{Hs}(X)| \leq C \cdot 2^{\frac{H}{2}} = C\sqrt{K}$ for some $C > 0$, hence $\hat{f}_{Hs}(X)$ is sub-exponential with parameters $(\sqrt{C^2 K}, 0)$.

### D.2.2. VERIFICATION OF ASSUMPTION 5.1

For any $s = 1, ..., k$, $z_s = \pm 1$ and $x = (t, y)$ with $t \in [\frac{s-1}{k}, \frac{s}{k}]$, we have

$$
p_{z^k}(x) = \frac{1}{k} \cdot k \left[ (\frac{1}{2} + \epsilon k^{-(r+\frac{1}{2})} z_s \psi_s^k(t)) \mathbb{1}_{y=1} + (\frac{1}{2} - \epsilon k^{-(r+\frac{1}{2})} z_s \psi_s^k(t)) \mathbb{1}_{y=0} \right].
$$

Note that

$$
\int_{\frac{s-1}{k}}^{\frac{s}{k}} \sum_{y \in \{0,1\}} k \left[ (\frac{1}{2} + \epsilon k^{-(r+\frac{1}{2})} z_s \psi_s^k(t)) \mathbb{1}_{y=1} + (\frac{1}{2} - \epsilon k^{-(r+\frac{1}{2})} z_s \psi_s^k(t)) \mathbb{1}_{y=0} \right] dt = 1,
$$

Hence $p_{s,z_s}(x) = k \left[ (\frac{1}{2} + \epsilon k^{-(r+\frac{1}{2})} z_s \psi_s^k(t)) \mathbb{1}_{y=1} + (\frac{1}{2} - \epsilon k^{-(r+\frac{1}{2})} z_s \psi_s^k(t)) \mathbb{1}_{y=0} \right] > 0$ is a distribution function on $[\frac{s-1}{k}, \frac{s}{k}] \times \{0, 1\}$.

### D.2.3. VERIFICATION OF ASSUMPTION 5.2

For any $s = 1, ..., k$, $z_s = \pm 1$ and $x = (t, y)$ with $t \in [\frac{s-1}{k}, \frac{s}{k}]$,

$$
\begin{aligned}
L_{s,z_s}(x) &= \log \left( \frac{(\frac{1}{2} - \epsilon k^{-(r+\frac{1}{2})} z_s \psi_s^k(t)) \mathbb{1}_{y=1} + (\frac{1}{2} + \epsilon k^{-(r+\frac{1}{2})} z_s \psi_s^k(t)) \mathbb{1}_{y=0}}{(\frac{1}{2} + \epsilon k^{-(r+\frac{1}{2})} z_s \psi_s^k(t)) \mathbb{1}_{y=1} + (\frac{1}{2} - \epsilon k^{-(r+\frac{1}{2})} z_s \psi_s^k(t)) \mathbb{1}_{y=0}} \right) \\
&= \log \left( 1 + \frac{2\epsilon k^{-(r+\frac{1}{2})} z_s \psi_s^k(t)(1 - 2\mathbb{1}_{y=1})}{(\frac{1}{2} + \epsilon k^{-(r+\frac{1}{2})} z_s \psi_s^k(t)) \mathbb{1}_{y=1} + (\frac{1}{2} - \epsilon k^{-(r+\frac{1}{2})} z_s \psi_s^k(t)) \mathbb{1}_{y=0}} \right).
\end{aligned}
$$

Then for $X = (T, Y) \sim p_{s,z_s}$, by 1) in Lemma 3.1, $|L_{s,z_s}(X)| \leq C' k^{-r}$ for some $C' > 0$, hence $L_{s,z_s}(X)$ is sub-exponential with parameters $(\sqrt{C^2 k^{-2r}}, 0)$. Moreover, by the inequality $-x - x^2 \leq \log(1 - x) \leq -x$ for $|x| < \frac{1}{2}$,

then

$$|\mathbb{E}[L_{s,z_s}(X)]| \leq \left| \mathbb{E}\left[ \frac{2\epsilon k^{-(r+\frac{1}{2})}z_s\psi_s^k(T)(1-2\mathbb{1}_{Y=1})}{(\frac{1}{2}+\epsilon k^{-(r+\frac{1}{2})}z_s\psi_s^k(T))\mathbb{1}_{Y=1} + (\frac{1}{2}-\epsilon k^{-(r+\frac{1}{2})}z_s\psi_s^k(T))\mathbb{1}_{Y=0}} \right] \right|$$

$$+ \mathbb{E}\left[ \left( \frac{2\epsilon k^{-(r+\frac{1}{2})}z_s\psi_s^k(T)(1-2\mathbb{1}_{Y=1})}{(\frac{1}{2}+\epsilon k^{-(r+\frac{1}{2})}z_s\psi_s^k(T))\mathbb{1}_{Y=1} + (\frac{1}{2}-\epsilon k^{-(r+\frac{1}{2})}z_s\psi_s^k(T))\mathbb{1}_{Y=0}} \right)^2 \right]$$

$$\preceq 0 + k^{-(2r+1)} \cdot 2^h \asymp k^{-2r},$$

completing the proof.

## 3. Nonparametric Poisson Regression Problem in Example 2.10

### D.2.1. VERIFICATION OF ASSUMPTION 4.1

Let $h(y) = y$, then $\hat{f}_{Hs}(X) = \phi_{Hs}(T)Y$. Note that

$$\mathbb{E}[\hat{f}_{Hs}(X)] = \mathbb{E}[\phi_{Hs}(T)\mathbb{E}[Y|T]] = \mathbb{E}[\phi_{Hs}(T)f(T)] = \int_0^1 f(t)\phi_{Hs}(t)dt = f_{Hs},$$

hence $\hat{f}_{Hs}(X)$ is an unbiased estimator of $f_{Hs}$.

By 1) in Lemma 3.1, we have $|\phi_{Hs}(T)| \leq C \cdot 2^{\frac{H}{2}} = C\sqrt{K}$ for some $C > 0$. Then by 4) in Lemma 3.1, $|\phi_{Hs}(T)f(T)| \leq CL'\sqrt{K}$ almost surely, hence $\phi_{Hs}(T)f(T)$ is sub-Gaussian with parameter $CL'\sqrt{K}$. Since $Y|T \sim \text{Poisson}(f(T))$, we have $\mathbb{E}\left[ e^{\lambda\phi_{Hs}(T)(Y-f(T))} \middle| T \right] = e^{f(T)(e^{\lambda\phi_{Hs}(T)} - \lambda\phi_{Hs}(T) - 1)}, \forall\lambda \in \mathbb{R}$. For any $\lambda$ with $|\lambda| < \frac{1}{C\sqrt{K}}$, we have $|\lambda\phi_{Hs}(T)| \leq 1$ and then $e^{\lambda\phi_{Hs}(T)} - \lambda\phi_{Hs}(T) - 1 \leq (\lambda\phi_{Hs}(T))^2$. Then

$$\mathbb{E}\left[ e^{\lambda(\hat{f}_{Hs}(X)-f_{Hs})} \right] = \mathbb{E}\left[ \mathbb{E}\left[ e^{\lambda\phi_{Hs}(T)(Y-f(T))} \middle| T \right] e^{\lambda(\phi_{Hs}(T)f(T)-f_{Hs})} \right]$$

$$\leq \mathbb{E}\left[ e^{(\lambda\phi_{Hs}(T))^2} e^{\lambda(\phi_{Hs}(T)f(T)-f_{Hs})} \right]$$

$$\leq e^{C^2K\lambda^2}\mathbb{E}\left[ e^{\lambda(\phi_{Hs}(T)f(T)-f_{Hs})} \right]$$

$$\leq e^{\frac{1}{2}C^2K(2+L'^2)\lambda^2},$$

implying that $\hat{f}_{Hs}(X)$ is sub-exponential with parameters $\left( \sqrt{C^2K(2+L'^2)}, C\sqrt{K} \right)$.

### D.2.2. VERIFICATION OF ASSUMPTION 5.1

For any $s = 1, ..., k$, $z_s = \pm 1$ and $x = (t, y)$ with $t \in [\frac{s-1}{k}, \frac{s}{k}]$, we have

$$p_{z^k}(x) = \frac{1}{k} \cdot ke^{-(C_0+\epsilon k^{-(r+\frac{1}{2})}z_s\psi_s^k(t))} \frac{(C_0 + \epsilon k^{-(r+\frac{1}{2})}z_s\psi_s^k(t))^y}{y!}.$$

Note that

$$\int_{\frac{s-1}{k}}^{\frac{s}{k}} \sum_{y\in\mathbb{N}} ke^{-(C_0+\epsilon k^{-(r+\frac{1}{2})}z_s\psi_s^k(t))} \frac{(C_0 + \epsilon k^{-(r+\frac{1}{2})}z_s\psi_s^k(t))^y}{y!}dt = 1,$$

Hence $p_{s,z_s}(x) = ke^{-(C_0+\epsilon k^{-(r+\frac{1}{2})}z_s\psi_s^k(t))} \frac{(C_0+\epsilon k^{-(r+\frac{1}{2})}z_s\psi_s^k(t))^y}{y!} > 0$ is a distribution function on $[\frac{s-1}{k}, \frac{s}{k}] \times \mathbb{N}$.

### D.2.3. VERIFICATION OF ASSUMPTION 5.2

For any $s = 1, ..., k$, $z_s = \pm 1$ and $x = (t, y)$ with $t \in [\frac{s-1}{k}, \frac{s}{k}]$,

$$L_{s,z_s}(x) = 2\epsilon k^{-(r+\frac{1}{2})}z_s\psi_s^k(t) + y\log\left( \frac{C_0 - \epsilon k^{-(r+\frac{1}{2})}z_s\psi_s^k(t)}{C_0 + \epsilon k^{-(r+\frac{1}{2})}z_s\psi_s^k(t)} \right).$$

Now let $X = (T, Y) \sim p_{s, z_s}$, then

$$L_{s, z_s}(X) = 2\epsilon k^{-(r+\frac{1}{2})} z_s \psi_s^k(T) + Y \log \left( 1 - \frac{2\epsilon k^{-(r+\frac{1}{2})} z_s \psi_s^k(T)}{C_0 + \epsilon k^{-(r+\frac{1}{2})} z_s \psi_s^k(T)} \right).$$

By 1) in Lemma 3.1, we have

$$\left| \log \left( 1 - \frac{2\epsilon k^{-(r+\frac{1}{2})} z_s \psi_s^k(T)}{C_0 + \epsilon k^{-(r+\frac{1}{2})} z_s \psi_s^k(T)} \right) \right| \leq \left| \frac{4\epsilon k^{-(r+\frac{1}{2})} z_s \psi_s^k(T)}{C_0 + \epsilon k^{-(r+\frac{1}{2})} z_s \psi_s^k(T)} \right| \leq C_1 k^{-r}$$

for some $C_1 > 0$. Then similar to the verification of Assumption 4.1, $Y \log \left( 1 - \frac{2\epsilon k^{-(r+\frac{1}{2})} z_s \psi_s^k(T)}{C_0 + \epsilon k^{-(r+\frac{1}{2})} z_s \psi_s^k(T)} \right)$ is sub-exponential with parameters $(\sqrt{C_1^2 k^{-2r}(2 + L'^2)}, C_1 k^{-r})$. By 1) in Lemma 3.1, $|2\epsilon k^{-(r+\frac{1}{2})} z_s \psi_s^k(T)| \leq C_2 k^{-r}$ for some $C_2 > 0$, hence $2\epsilon k^{-(r+\frac{1}{2})} z_s \psi_s^k(T)$ is sub-exponential with parameters $(\sqrt{C_2^2 k^{-2r}}, 0)$. Therefore $L_{s, z_s}(X)$ is sub-exponential with parameters $(\sqrt{[C_1^2(2 + L'^2) + C_2^2] k^{-2r}}, C_1 k^{-r})$ (by the discussion in Appendix A.1).

Moreover, note that

$$|\mathbb{E}[L_{s, z_s}(X)]| = \left| 2\epsilon k^{-(r+\frac{1}{2})} z_s \mathbb{E}\left[\psi_s^k(T)\right] + \mathbb{E}\left[ \mathbb{E}[Y|T] \log \left( \frac{C_0 - \epsilon k^{-(r+\frac{1}{2})} z_s \psi_s^k(T)}{C_0 + \epsilon k^{-(r+\frac{1}{2})} z_s \psi_s^k(T)} \right) \right] \right|$$

$$= \left| \mathbb{E}\left[ f(T) \log \left( \frac{1 - C_0^{-1} \epsilon k^{-(r+\frac{1}{2})} z_s \psi_s^k(T)}{1 + C_0^{-1} \epsilon k^{-(r+\frac{1}{2})} z_s \psi_s^k(T)} \right) \right] \right|.$$

Since $f(T) > 0$, by the inequality $-2x - 4x^2 \leq \log \frac{1-x}{1+x} \leq -2x + 4x^2$ for $|x| < \frac{1}{2}$,

$$|\mathbb{E}[L_{s, z_s}(X)]| \leq 2 \left| \mathbb{E}\left[ f(T) C_0^{-1} \epsilon k^{-(r+\frac{1}{2})} z_s \psi_s^k(T) \right] \right| + 4\mathbb{E}\left[ f(T) \left( C_0^{-1} \epsilon k^{-(r+\frac{1}{2})} z_s \psi_s^k(T) \right)^2 \right]$$

$$\leq 2C_0^{-1} \epsilon k^{-(r+\frac{1}{2})} f_{h - h_0, s} + 4L' \left( C_0^{-1} \epsilon k^{-(r+\frac{1}{2})} \right)^2 \cdot k$$

$$\preceq k^{-(r+\frac{1}{2})} \|f - f^{h - h_0 - 1}\|_2 + k^{-2r} \preceq k^{-2r},$$

where the second inequality is by 1) in Lemma 3.1 and the last inequality is by 3) in Lemma 3.1. This completes the proof.

## 4. Nonparametric Heteroskedastic Regression Problem in Example 2.11

### D.2.1. VERIFICATION OF ASSUMPTION 4.1

Let $h(y) = y^2$, then $\hat{f}_{Hs}(X) = \phi_{Hs}(T) Y^2$. Note that

$$\mathbb{E}[\hat{f}_{Hs}(X)] = \mathbb{E}[\phi_{Hs}(T) \mathbb{E}[Y^2|T]] = \mathbb{E}[\phi_{Hs}(T) f(T)] = \int_0^1 f(t) \phi_{Hs}(t) dt = f_{Hs},$$

hence $\hat{f}_{Hs}(X)$ is an unbiased estimator of $f_{Hs}$.

By 1) in Lemma 3.1, we have $|\phi_{Hs}(T)| \leq C \cdot 2^{\frac{H}{2}} = C\sqrt{K}$ for some $C > 0$. Then by 4) in Lemma 3.1, $|\phi_{Hs}(T) f(T)| \leq CL'\sqrt{K}$ almost surely, hence $\phi_{Hs}(T) f(T)$ is sub-Gaussian with parameter $CL'\sqrt{K}$. Since $Y^2|T \sim f(T)\chi_1^2$, we have $\mathbb{E}\left[ e^{\lambda \phi_{Hs}(T)(Y^2 - f(T))} \Big| T \right] = \frac{e^{-\lambda \phi_{Hs}(T) f(T)}}{\sqrt{1 - 2\lambda \phi_{Hs}(T) f(T)}} \leq e^{2(\phi_{Hs}(T) f(T) \lambda)^2}, \forall |\lambda| < \frac{1}{4CL'\sqrt{K}}$, where the last inequality is by $e^{-x - 2x^2} \leq \sqrt{1 - 2x}$ for $|x| < \frac{1}{4}$. For any $\lambda$ with $|\lambda| < \frac{1}{4CL'\sqrt{K}}$,

$$\mathbb{E}\left[ e^{\lambda(\hat{f}_{Hs}(X) - f_{Hs})} \right] = \mathbb{E}\left[ \mathbb{E}\left[ e^{\lambda \phi_{Hs}(T)(Y^2 - f(T))} \Big| T \right] e^{\lambda(\phi_{Hs}(T) f(T) - f_{Hs})} \right]$$

$$\leq \mathbb{E}\left[ e^{2(\lambda \phi_{Hs}(T) f(T))^2} e^{\lambda(\phi_{Hs}(T) f(T) - f_{Hs})} \right]$$

$$\leq e^{2C^2 K L'^2 \lambda^2} \mathbb{E}\left[ e^{\lambda(\phi_{Hs}(T) f(T) - f_{Hs})} \right]$$

$$\leq e^{\frac{1}{2} \cdot 5C^2 K L'^2 \lambda^2},$$

implying that $\hat{f}_{Hs}(X)$ is sub-exponential with parameters $(\sqrt{5C^2 K L'^2}, 4CL'\sqrt{K})$.

D.2.2. VERIFICATION OF ASSUMPTION 5.1

For any $s = 1, ..., k$, $z_s = \pm 1$ and $x = (t, y)$ with $t \in [\frac{s-1}{k}, \frac{s}{k}]$, we have

$$
p_{z^k}(x) = \frac{1}{k} \cdot \frac{k}{\sqrt{2\pi \left(C_0 + \epsilon k^{-(r+\frac{1}{2})} z_s \psi_s^k(t)\right)}} e^{-\frac{y^2}{2\left(C_0 + \epsilon k^{-(r+\frac{1}{2})} z_s \psi_s^k(t)\right)}}.
$$

Note that

$$
\int_{\frac{s-1}{k}}^{\frac{s}{k}} \int_{\mathbb{R}} \frac{k}{\sqrt{2\pi \left(C_0 + \epsilon k^{-(r+\frac{1}{2})} z_s \psi_s^k(t)\right)}} e^{-\frac{y^2}{2\left(C_0 + \epsilon k^{-(r+\frac{1}{2})} z_s \psi_s^k(t)\right)}} \, dy \, dt = 1,
$$

Hence $p_{s, z_s}(x) = \frac{k}{\sqrt{2\pi \left(C_0 + \epsilon k^{-(r+\frac{1}{2})} z_s \psi_s^k(t)\right)}} e^{-\frac{y^2}{2\left(C_0 + \epsilon k^{-(r+\frac{1}{2})} z_s \psi_s^k(t)\right)}} > 0$ is a distribution function on $[\frac{s-1}{k}, \frac{s}{k}] \times \mathbb{R}$.

D.2.3. VERIFICATION OF ASSUMPTION 5.2

For any $s = 1, ..., k$, $z_s = \pm 1$ and $x = (t, y)$ with $t \in [\frac{s-1}{k}, \frac{s}{k}]$,

$$
L_{s, z_s}(x) = \frac{1}{2} \log \left( \frac{C_0 - \epsilon k^{-(r+\frac{1}{2})} z_s \psi_s^k(t)}{C_0 + \epsilon k^{-(r+\frac{1}{2})} z_s \psi_s^k(t)} \right) + \frac{y^2}{2} \left( \frac{1}{C_0 - \epsilon k^{-(r+\frac{1}{2})} z_s \psi_s^k(t)} - \frac{1}{C_0 + \epsilon k^{-(r+\frac{1}{2})} z_s \psi_s^k(t)} \right).
$$

Now let $X = (T, Y) \sim p_{s, z_s}$, then

$$
L_{s, z_s}(X) = \frac{1}{2} \log \left( 1 - \frac{2\epsilon k^{-(r+\frac{1}{2})} z_s \psi_s^k(T)}{C_0 + \epsilon k^{-(r+\frac{1}{2})} z_s \psi_s^k(T)} \right) + Y^2 \cdot \frac{\epsilon k^{-(r+\frac{1}{2})} z_s \psi_s^k(T)}{C_0^2 - (\epsilon k^{-(r+\frac{1}{2})} z_s \psi_s^k(T))^2}.
$$

By 1) in Lemma 3.1, we have

$$
\left| \frac{1}{2} \log \left( 1 - \frac{2\epsilon k^{-(r+\frac{1}{2})} z_s \psi_s^k(T)}{C_0 + \epsilon k^{-(r+\frac{1}{2})} z_s \psi_s^k(T)} \right) \right| \leq \left| \frac{2\epsilon k^{-(r+\frac{1}{2})} z_s \psi_s^k(T)}{C_0 + \epsilon k^{-(r+\frac{1}{2})} z_s \psi_s^k(T)} \right| \leq C_1 k^{-r}
$$

for some $C_1 > 0$. Hence $\frac{1}{2} \log \left( 1 - \frac{2\epsilon k^{-(r+\frac{1}{2})} z_s \psi_s^k(T)}{C_0 + \epsilon k^{-(r+\frac{1}{2})} z_s \psi_s^k(T)} \right)$ is sub-exponential with parameters $(\sqrt{C_1^2 k^{-2r}}, 0)$. By 1) in Lemma 3.1, we have

$$
\left| \frac{\epsilon k^{-(r+\frac{1}{2})} z_s \psi_s^k(T)}{C_0^2 - (\epsilon k^{-(r+\frac{1}{2})} z_s \psi_s^k(T))^2} \right| \leq C_2 k^{-r}.
$$

Then similar to the verification of Assumption 4.1, $Y^2 \cdot \frac{\epsilon k^{-(r+\frac{1}{2})} z_s \psi_s^k(T)}{C_0^2 - (\epsilon k^{-(r+\frac{1}{2})} z_s \psi_s^k(T))^2}$ is sub-exponential with parameters $(\sqrt{5 C_2^2 k^{-2r} L'^2}, 4 C_2 L' k^{-r})$. Therefore $L_{s, z_s}(X)$ is sub-exponential with parameters $(\sqrt{(C_1^2 + 5 C_2^2 L'^2) k^{-2r}}, 4 C_2 L' k^{-r})$ (by the discussion in Appendix A.1).

Moreover, note that

$$
|\mathbb{E}[L_{s, z_s}(X)]| = \left| \frac{1}{2} \mathbb{E} \left[ \log \left( \frac{C_0 - \epsilon k^{-(r+\frac{1}{2})} z_s \psi_s^k(t)}{C_0 + \epsilon k^{-(r+\frac{1}{2})} z_s \psi_s^k(t)} \right) \right] + \mathbb{E} \left[ \mathbb{E}[Y^2 | T] \cdot \frac{\epsilon k^{-(r+\frac{1}{2})} z_s \psi_s^k(T)}{C_0^2 - (\epsilon k^{-(r+\frac{1}{2})} z_s \psi_s^k(T))^2} \right] \right|
$$

$$
\leq \frac{1}{2} \left| \mathbb{E} \left[ \log \left( \frac{1 - C_0^{-1} \epsilon k^{-(r+\frac{1}{2})} z_s \psi_s^k(t)}{1 + C_0^{-1} \epsilon k^{-(r+\frac{1}{2})} z_s \psi_s^k(t)} \right) \right] \right| + \left| \mathbb{E} \left[ f(T) \cdot \frac{\epsilon k^{-(r+\frac{1}{2})} z_s \psi_s^k(T)}{C_0^2 - (\epsilon k^{-(r+\frac{1}{2})} z_s \psi_s^k(T))^2} \right] \right|.
$$

For the first term, by the inequality $-2x - 4x^2 \leq \log \frac{1-x}{1+x} \leq -2x + 4x^2$ for $|x| < \frac{1}{2}$,

$$\left| \mathbb{E}\left[ \log\left( \frac{1 - C_0^{-1} \epsilon k^{-(r+\frac{1}{2})} z_s \psi_s^k(t)}{1 + C_0^{-1} \epsilon k^{-(r+\frac{1}{2})} z_s \psi_s^k(t)} \right) \right] \right|$$

$$\leq 2 \left| \mathbb{E}\left[ C_0^{-1} \epsilon k^{-(r+\frac{1}{2})} z_s \psi_s^k(T) \right] \right| + 4 \mathbb{E}\left[ \left( C_0^{-1} \epsilon k^{-(r+\frac{1}{2})} z_s \psi_s^k(T) \right)^2 \right]$$

$$\preceq 0 + k^{-2r} = k^{-2r}.$$

For the second term, since $f(T) > 0$, then by the inequality $x - x^2 \leq \frac{x}{1-x^2} \leq x + x^2$ for $|x| < \frac{1}{2}$,

$$\left| \mathbb{E}\left[ f(T) \cdot \frac{\epsilon k^{-(r+\frac{1}{2})} z_s \psi_s^k(T)}{C_0^2 - (\epsilon k^{-(r+\frac{1}{2})} z_s \psi_s^k(T))^2} \right] \right|$$

$$\leq C_0^{-1} \left| \mathbb{E}\left[ f(T) C_0^{-1} \epsilon k^{-(r+\frac{1}{2})} z_s \psi_s^k(T) \right] \right| + C_0^{-1} \mathbb{E}\left[ f(T) \left( C_0^{-1} \epsilon k^{-(r+\frac{1}{2})} z_s \psi_s^k(T) \right)^2 \right]$$

$$\preceq k^{-(r+\frac{1}{2})} \|f - f^{h-h_0-1}\|_2 + k^{-2r} \preceq k^{-2r},$$

where the second inequality is by 1) in Lemma 3.1 and the last inequality is by 3) in Lemma 3.1. Combining these two parts yields $|\mathbb{E}[L_{s,z_s}(X)]| \preceq k^{-2r}$, which completes the proof.

