# OpenReview forum: "Distributed Nonparametric Estimation: from Sparse to Dense Samples per Terminal"
_ICML.cc/2025/Conference — ICML 2025 poster_

### Official Review · Reviewer_tqL9 · 2025-03-15

**Overall Recommendation:** 4

**Summary:**

This paper studies nonparametric function estimation under communication constraints, where each distributed terminal holds multiple i.i.d. samples and can communicate sequentially. The authors establish nearly minimax optimal rates across different regimes and identify phase transitions as the number of samples per terminal varies from sparse to dense. This work extends previous studies, which were limited to either dense-sample settings or a single sample per terminal.

To achieve the optimal rates, the authors propose a layered estimation protocol that builds on parametric density estimation techniques. The proposed two-phase scheme first employes wavelet transform to convert the nonparametric density estimation problem into a parametric one, and then solve it using quantization and other sample assignment tricks. The upper bounds are then complemented with information-theoretic lower bounds based on SDPIs and Assouad-typed inequalities. The results apply to various special cases, including density estimation and regression models with Gaussian, binary, Poisson, and heteroskedastic noise.

**Claims And Evidence:**

The main theorems appear to be correct to my best knowledge.

**Essential References Not Discussed:**

The references are adequate to my best knowledge.

**Experimental Designs Or Analyses:**

N/A since this is a theory paper.

**Methods And Evaluation Criteria:**

N/A since this is a theory paper.

**Other Comments Or Suggestions:**

See above.

**Other Strengths And Weaknesses:**

This work makes a significant contribution to the study of distributed nonparametric estimation under communication constraints by addressing multiple sample regimes. It successfully closes gaps left by previous research, particularly by characterizing minimax optimal rates across different data regimes and identifying phase transitions in estimation performance. The theoretical insights and proposed techniques offer valuable advancements in this area.

Overall, I did not identify any major weaknesses in the paper. The results appear sound (although the upper bounds somewhat heavily rely on the prior work [Yuan et al. 2024]), and the contributions are well-motivated. However, I have a few suggestions to improve the clarity and presentation of the work:

- **Reorganization of the Introduction and Literature Review:** The first two sections could be slightly restructured, as the literature review is spread across Sections 1.2 and 2.2. Integrating these discussions into a more cohesive review of prior work would improve readability and make it easier for readers to understand the context and contributions of this paper.

- **Clarifying the Technical Contributions:** The proposed upper and lower bounds are derived using different techniques tailored to specific data regimes. The authors could better emphasize these techniques and compare them explicitly with prior methods, which often focus only on particular regimes. This would help highlight the novelty of the approach and provide insights into which aspects of the methodology are new and which build on existing work.

- **Potential Broader Impact:** Given the generality of the techniques used, the authors might briefly discuss whether their methods could extend to other communication-constrained estimation problems (e.g., other parametric models) beyond the specific settings considered in the paper. Also, does the similar techniques apply to local differential privacy constraints, which are often studied together with the communication constraints?

**Questions For Authors:**

The paper considers a sequentially interactive communication model, with the upper bound based on [Yuan et al. 2024], which relies on an adaptive refinement protocol requiring sequential interaction. Do the authors believe sequential interaction is essential? Could a similar result be achieved under an independent communication model?

Additionally, do the lower bound techniques extend to the blackboard communication model, which is more general than the sequential model?

**Relation To Broader Scientific Literature:**

This work closes the gap between distributed nonparametric estimation under communication constraints with multiple sample regimes. Similar techniques may be applied to distributed estimation under privacy constraints as well.

**Theoretical Claims:**

The main theorems, including the proposed schemes and the lower bounds, appear to be correct to my best knowledge.

---

> ### Author Rebuttal · Authors · 2025-03-29
>
> Thanks for your detailed reading and acknowledging the generality of our technique. We will carefully revise our paper according to your suggestions. In the following we provide responses to your questions.
>
> ## Responses to the suggestions:
>
> -   **Reorganization of the Introduction and Literature Review.**
>
> It is a good suggestion. We will reorganize Subsections 1.2 and 2.2 to improve the presentation.
>
> -   **Clarifying the Technical Contributions.**
>
> We will follow your suggestion and add more discussions to clarify the technical contributions over previous works. Specifically, In Section 4 Remark 4.3, 4.5 about the comparisons of techniques for proving the upper bounds will be highlighted. Parts of Remark C.4 and C.5 will be moved to be Section 5 of main body, to clarify our technical novelty for the proof of the lower bounds.
>
> -   **Potential Broader Impact.**
>
> It is a good suggestion. We will add a conclusion and discussion section at the end of the revised paper to provide some potential directions with the help of our methods, especially in the privacy-constrained problems.
>
> ## Responses to the questions:
>
> -   **Do the authors believe sequential interaction is essential? Could a similar result be achieved under an independent communication model?**
>
> It may be difficult to design an independent communication model for all regimes discussed in this work. For some regimes it is possible, especially in the case where $l$ is relatively larger. But for others, especially for the regime where the optimal rate depends exponentially on the number of communication bits, it is much more difficult. In light of our two-layer protocol, the solution may depend on an independent protocol for the parametric distribution estimation problem in [Yuan et al. 2024]. But for the latter, we still do not have ideas, especially for the corresponding regime where the optimal rate depends exponentially on the communication budgets.
>
> -   **Additionally, do the lower bound techniques extend to the blackboard communication model, which is more general than the sequential model?**
>
> It is a good question and we will think about it more carefully. We think it is possible, but it may need some major improvements in the proof techniques.

---

### Official Review · Reviewer_nezZ · 2025-03-16

**Overall Recommendation:** 3

**Summary:**

This paper investigates the phase transition in optimal estimation rates as the number of samples per terminal increases from sparse to dense in a distributed setting. The results are purely theoretical, filling gaps in the existing literature and offering a wide range of applications. Overall, the paper is well-written, with a clear and coherent presentation.

**Claims And Evidence:**

Yes.

**Essential References Not Discussed:**

No.

**Experimental Designs Or Analyses:**

There is no experiment.

**Methods And Evaluation Criteria:**

There is no dataset and no empirical study in the paper.

**Other Comments Or Suggestions:**

All the examples in Section 2.3 appear quite simple, focusing on nonparametric estimation of a one-dimensional mean, variance, or density function. Such one-dimensional problems are way too simple compared to other works appear in ICML.  One indication of this simplicity is that each estimated wavelet coefficient, $\hat f_{Hs}$ , has a straightforward closed-form expression, which naturally simplifies the theoretical analysis. However, the practical motivation for this estimation protocol is not entirely clear. In my view, a stronger justification from real-world applications is needed to highlight the importance of such a distributed estimation procedure.

**Other Strengths And Weaknesses:**

Strength: the paper is well-written and the theoretical results make sense.
Weakness: the problem under investigation seems a bit simple and there is no strong motivation from the practical applications. There is no numerical investigations to support the theoretical findings.

**Questions For Authors:**

1. What is the intuition behind the effective sample size $N_{ess}$ given in (2)? Providing such intuition would help readers better understand the phase transition.

2. In lines 165–167, I don’t fully understand why it is considered impractical to require $N_{ess} > n$. If $N_{ess} < n$, one could simply use data from a single terminal, eliminating communication costs altogether. This approach would also yield a faster convergence rate of $n^{-\frac{2r}{2r+1}}$ compared to the rate given in the paper, $N_{ess}^{-\frac{2r}{2r+1}}$. Could you clarify this point?

3. What is the specific rate for $H$ in each transition phase? The resolution $H$ is essentially a tuning parameter that controls the trade off between the smoothness of the estimated function and the goodness of fit to the data. And the choice of $H$ also affects the communication cost as a larger $H$ indicates more data to be communicated.

4. Related to the previous question, how do you choose the optimal resolution $H$? For the tuning parameter $\lambda$ in the distributed kernel ridge regression,  the one needs to intentionally under-smooth the function estimator in each terminal (by using a sub-optimal $\lambda$ ) so that the aggregated function is globally optimal, see, e.g., [1], [2], [3]. In particular, [2] and [3] propose a GCV criterion to empirically choose the best tuning parameter. I suspect that the choice of $H$ in this paper also has similar phenomena, and can you provide some discussion on this important issue?

5. There is no difference between Theorem 2.12 and 2.11, why use two theorems?

6. Consider the same problem studied in Zhu & Lafferty (2018). The work of \cite{ref4} proposes a different quantization scheme that does not depend on the Fourier coefficients of the underlying function. It would be beneficial to include this in the literature review for completeness.

7. It would be more convincing if some numerical experiments were provided to validate the theoretical findings in the paper.


References:

[1]. Zhang, Y., Duchi, J., & Wainwright, M. (2015). Divide and conquer kernel ridge regression: A distributed algorithm with minimax optimal rates. The Journal of Machine Learning Research, 16(1), 3299-3340.

[2]. Xu, G., Shang, Z., & Cheng, G. (2018). Optimal tuning for divide-and-conquer kernel ridge regression with massive data. In International Conference on Machine Learning (pp. 5483-5491). PMLR.

[3]. Xu, G., Shang, Z., & Cheng, G. (2019). Distributed generalized cross-validation for divide-and-conquer kernel ridge regression and its asymptotic optimality. Journal of computational and graphical statistics, 28(4), 891-908.

[4]. Li, K., Liu, R., Xu, G., & Shang, Z. (2024). Nonparametric Inference under B-bits Quantization. Journal of Machine Learning Research, 25(19), 1-68.

**Relation To Broader Scientific Literature:**

Fill in the theoretical gaps in the literature.

**Theoretical Claims:**

I did not check the proof line by line, but the arguments make intuitive sense.

---

> ### Author Rebuttal · Authors · 2025-03-29
>
> Thanks for your careful reading and interesting questions. Here are our detailed responses.
>
> ## Responses to the weaknesses:
>
> -   **The problem under investigation seems a bit simple. Such one-dimensional problems are way too simple.** $\hat{f}_{Hs}$ **has a straightforward closed-form expression. ...**
>
> The model is idealized compared with practical applications. However, there are enough motivations to study it, and we will highlight them in the revised paper.
>
> 1.  Our formulation does contain many important statistical estimation problems described in Section 2.3. Assumption 4.1 on the existence of the estimated wavelet coefficient $\hat{f}_{Hs}$ is satisfied by many specific problems with i.i.d. random generated sample, including density estimation and several common nonparametric regression problems with random design. None of these problems are trivial.
> 2.  We assume that the function to be estimated is one-dimensional, but the problem is complex since the function space is infinite-dimensional. As a result, nonparametric function estimation problem can be seen as a useful model of practical applications, providing a theoretical perspective to understand the difficulty of distributed learning.
> 3.  Our common goal is to make the model closer to the practical case. This work is only one step towards the goal, but not its end.
>
> ## Responses to the questions:
>
> -   **What is the intuition behind the effective sample size given in (2)?**
>
> The intuition behind the definition of $N_{ess}$ can be obtained by making comparisons between cases with and without communication constraints. If there are no communication constraints, the optimal rate is $N^{-\frac{2r}{2r+1}}$. In this work we show that the effect of the communication constraints seems to reduce the sample size from $N$ to $N_{ess}$. The optimal rate for the problem is then roughly $N_{ess}^{-\frac{2r}{2r+1}}$.
>
> -   **Why is it considered impractical to require** $N_{ess}>n$?
>
> It may be impractical in some real-world system, where it is impossible to put the central decider together with each of the distributed terminals to process the raw data. For example, imagine the case where the terminal is a small unmanned aerial vehicle short of electricity. Since communication is necessary, it is possible that $N_{ess} < n$, where the strict communication constraints severely affect the performance of the system.
>
> -   **What is the specific rate for** $H$ **in each transition phase? ... For the tuning parameter** $\lambda$ **in the distributed kernel ridge regression, see, e.g., [1], [2], [3]. ...**
>
> Since these two questions are highly related, we respond to them together.
>
> 1.  In our estimation protocol, the choice of the resolution $H$ is explicitly given by Equation (11) for each parametric regime in the preparation phase in Section 4.1.
>
> 2.  Your understanding of the tuning parameter $H$ is right. Due to the communication constraints, the choice of $H$ by (11) can no longer achieve the optimal rate for the problem without communication constraints, namely $N^{-\frac{2r}{2r+1}}$. However, it is almost optimal for the communication constrained problem, in the sense that the resulting protocol can almost achieve the corresponding minimax optimal rate $N_{ess}^{-\frac{2r}{2r+1}}$.
>
> 3.  Note that the optimal $H$ has a closed-form expression by (11), as a function of parameters $(m,n,l,r)$. It is slightly different from the case considered in [1], [2], [3], where some of the system parameters may be absent, hence the optimal $\lambda$ is not a priori and adaptive data-driven method like cross-validation for tuning $\lambda$ is necessary. The works [1], [2], [3] also provide a good direction for our future work. Thank you for letting us know about them.
>
> -   **... no difference between Theorem 2.12 and 2.11...**
>
> I am a little bit confused with this question since 2.11 is an example but not a theorem. Do you want to ask about Theorems 2.1, 2.7 and 2.12? The difference is that Theorem 2.1 is for the general framework. In contrast, Theorem 2.7 is only for density estimation problems and Theorem 2.12 is only for regression problems. Both problems are subsumed by the general framework.
>
> -   **The work of [4] proposes a different quantization scheme that does not depend on the Fourier coefficients...**
>
> Thanks for letting us know about the fantastic work [4] and we will cite it in the revised version. In my understanding, it studies the nonparametric regression problems under fixed design, where the explanatory variables of the $n$ samples at each distributed side are located at $\frac{1}{n},\frac{2}{n}...,1$. It is interesting to compare this case with the random design setting (e.g. Examples 2.8-2.11) in our work.
>
> -   **... numerical experiments...**
>
> It is a good suggestion to add numerical experiments to highlight the theoretical contributions in future works. But for the current paper, due to the page limitation it may be hard to add such an entire section.

---

> > ### Comment · Reviewer_nezZ · 2025-04-05
> >
> > Thank you for the response!
> >
> > For 1, What is the intuition behind the effective sample size given in (2)? What I meant is that why the $N_{ess}$ is of the form given in (2), and what are the intuition behind specific forms in each case scenario.
> >
> > I would also suggest adding some discussion on the empirical choice of $H$ in the conclusion section.
> >
> > I would also suggesting adding some small scale experiments in Supplement.

---

> > > ### Author Response · Authors · 2025-04-08
> > >
> > > Thanks for your additional suggestions and the question. For your additional suggestions, we will prepare our revised paper according to them. Especially, in the conclusion section, we will also add discussions on the empirical choice of $H$ as a future direction, where related works such as [1], [2], [3] will be compared.
> > >
> > > -   For your first question, we have to confess that the specific forms of $N_{ess}$ for all cases may not be easy to explain in just a few words, since it involves much detailed computation. As we have pointed out, characterizing $N_{ess}$ is equivalent to characterizing the optimal rate $R(m,n,l,r)$. To make the intuition more clear, we believe that it may be better to describe how different cases in the optimal rate and $N_{ess}$ are divided. Below the basic understanding is provided from the perspective of the upper bound and the achievable protocol.
> > >
> > >     Take the density estimation problem for an example. We can approximate the density function in the Sobolev ball by constructing a histogram of samples at the decoder side. In this way, we can quantize the interval $[0,1]$ into $K = 2^H$ bins. The process of generating random samples is similar to throwing balls into bins, with each sample being a ball. Hence each terminal has $n$ balls and it can construct its own histogram. The problem is how should different terminals cooperate to send their messages, so that the decoder can construct a more detailed histogram.
> > >
> > >     There are two general strategies of encoding the histograms. The first strategy is by transmitting sequence numbers of the bins for balls. The second strategy is transmitting the number of balls in each bin. These two strategies are competing and we always choose one of them achieving a smaller error. It turns out that in Cases 1 and 2, the first strategy is better, while in Cases 3 and 4 the second strategy is better. For both strategies, as $l$ increases it is relatively easier to get a coarser estimate (Cases 1 and 3) at first and it becomes harder to make the estimate finer (Cases 2 and 4), until the best rate $(mn)^{-\frac{2r}{2r+1}}$ is obtained at last. So each strategies are split into two stages, and this explains the $[(\cdot \wedge \cdot) \vee (\cdot \wedge \cdot) ] \wedge mn$ structure of $N_{ess}$.
> > >
> > >     In each stage of each strategy, the parameter $K$, or equivalently $H$, is carefully chosen to minimize the estimation error (where details can be found in Section 4.2). Then for each specific case, the specific forms for the effective sample size $N_{ess}$ and the optimal rate are obtained.

---

### Official Review · Reviewer_yQDG · 2025-03-18

**Overall Recommendation:** 3

**Summary:**

The paper studies the problem of distributed nonparametric estimation under communication constraints, where each terminal holds multiple i.i.d. samples. The authors characterize the minimax optimal rates across all regimes, covering the transition from sparse to dense samples per terminal. They propose a two-layer estimation protocol that leverages parametric density estimation methods and wavelet-based estimators, deriving upper and lower bounds using information-theoretic techniques. The results extend the scope of existing works and apply to density estimation, Gaussian regression, and Poisson regression.

**Claims And Evidence:**

The claims in the submission are generally well-supported by comprehensive theoretical analysis.

**Essential References Not Discussed:**

I don't see additional references to add.

**Experimental Designs Or Analyses:**

It is not applicable here as the paper is primarily theoretical.

**Methods And Evaluation Criteria:**

The proposed algorithm utilizes ideas from recent advances by Acharya et al. (2020) and Yuan et al. (2024) and looks reasonable. However, I didn't find an experimental evaluation for the proposed algorithm, as the paper is primarily theoretical.

**Other Comments Or Suggestions:**

- Adding a conclusion section would improve the paper's structure.
- In the comparisons to prior works, please highlight how the paper's results and proofs utilize (or are inspired by) prior works, especially Acharya et al. (2020) and Yuan et al. (2024).
- Please also see the prior sections.

**Other Strengths And Weaknesses:**

Strengths:
1. The paper provides a complete characterization of the minimax rates for distributed nonparametric estimation, filling gaps left by previous studies.
2. The two-layer estimation protocol is well-structured and theoretically justified. The idea itself also possesses some novelty.
3. The paper leverages information-theoretic tools, which is a good plus.
4. The results generalize across various estimation problems, making the algorithm widely applicable.

Weaknesses:
1. While the results cover all regimes, some assumptions seem slightly more substantial than in prior works (e.g., Zaman & Szabó, 2022). Maybe some further discussion on the restrictiveness of these assumptions is beneficial.
2. The empirical justification is absent. While the work is primarily theoretical, a small-scale/synthetic experimental validation could illustrate the phase transitions in minimax rates.
3. The lower bound proof's technical novelty is somewhat limited, as it is unclear if the technique introduces fundamentally new proof strategies beyond adapting existing inequalities and/or constructions.
4. The discussion of practical communication constraints is somewhat idealized. In real-world distributed learning, bit quantization strategies often involve additional noise or compression losses. How does the model handle these imperfections?

**Questions For Authors:**

Please see the Strengths and Weaknesses section.

**Relation To Broader Scientific Literature:**

This paper extends prior work on distributed nonparametric estimation under communication constraints by fully characterizing minimax rates across all regimes, from sparse to dense samples per terminal. The two-layer estimation protocol builds on ideas from Acharya et al. (2024), who studied density estimation in the highly sparse case ($n=1$). Still, this work generalizes the approach to handle multiple samples per terminal.

**Theoretical Claims:**

I reviewed the proof sketches for the upper bound, which seemed reasonable. Due to time constraints, I did not check the detailed proofs in the appendix/supplementary.

---

> ### Author Rebuttal · Authors · 2025-03-29
>
> Thanks for your careful reading and detailed comments. Here are our point-to-point responses to your concerns.
>
> ## Responses to the weaknesses:
>
> -   **... some assumptions seem slightly more substantial than in prior works (e.g., Zaman & Szabó, 2022). ...**
>
> We want to clarify the rationalities of these assumptions here. We will make the comparisons more clear in the revised paper, by presenting more explanations about them immediately after the formulation of them.
>
> 1.  As you see, we want to address the general case not resolved by (Zaman & Szabo, 2022), which is substantially more difficult than some special cases.
> 2.  We want to make them easier to verify for specific models. Different from that in (Zaman & Szabo, 2022), all of our assumptions are imposed on single random variables. They also admit simpler expressions than that in (Zaman & Szabo, 2022).
> 3.  Even though stronger assumptions are imposed, our framework can also subsume almost all the specific models presented in (Zaman & Szabo, 2022).
>
> -   **The empirical justification is absent.**
>
> Adding synthetic experimental validation is a really good advice to highlight the theoretical contributions. But for the current paper, it seems that there may be little room for that. We will do such experiments in our future work.
>
> -   **The lower bound proof's technical novelty...**
>
> We give more explanations about our lower bound proof here. Since our contributions are mainly on the matching lower bounds for different parametric regimes, the novelty in the proof's technical novelty was not sufficiently highlighted. To overcome the technically more difficult setting in this work, we make two major improvements on the proof techniques.
>
> 1.  Stronger inequalities (eg. Lemmas 5.5 and 5.6) are proved and they are compared with previous methods in Remark C.4 and C.5. We apologize for not including these discussions in the main body due to the page limitation. In short, the boundedness or symmetry properties needed in previous methods are not satisfied for the problem considered here, and we develop effective inequalities to overcome the difficulty.
> 2.  We apply these stronger inequalities to new cases that have not been analyzed by previous works. The key to developing lower bounds for all these regimes is to fully characterize the differences in the structures of these regimes. The differences are revealed by analyzing the properties of the log-likelihood ratio defined in (20), shown especially in Section 5.2.2-5.2.3, by incorporating the balls-and-bins model. This is in contrast to previous works, where parametric regimes with $m \geq n^{2r} > 1$ are not considered and the structure of the problem in these regimes is not investigated.
>
> We will add more discussions in the revised paper and make our contributions on the lower bound more explicitly.
>
> -   **The discussion of practical communication constraints is somewhat idealized. ...**
>
> In real-world distributed learning, it is true that the communication between different agents may not use explicit bit quantization strategies. However, it is meaningful to model the practical distributed learning problem by the idealized one with bit quantization strategies. There are two reasons for that.
>
> 1.  The basic way of storing and sending messages by computers and any forms of digital communication systems is to encode them into bit strings. The messages may have various forms such as float numbers, but they are encoded into bit strings in their underlying implementation. In this sense, we do not lose too much with the simplified model.
>
> 2.  It may seem as if that bit quantization strategies often involve additional noise or compression losses with other ways of quantization like the floating number. But following the first point, as long as digital communications are used, our lower bounds show that the loss in estimation error is only up to logarithmic factors. In other words, the additional losses in the estimation error almost do not affect the convergence rates.
>
> I agree with you that there may be a gap between the idealized model and the practical case, but we hope that analysis of the idealized model can shed light on the understanding of the practical case.
>
> ## Responses to other comments or suggestions:
>
> -   **Adding a conclusion section...**
>
> We will follow this suggestion by adding a conclusion and discussion section at the end of the revised paper.
>
> -   **In the comparisons to prior works, please highlight how the paper's results and proofs utilize prior works especially Acharya et al. (2020) and Yuan et al. (2024).**
>
> We will revise Subsection 1.2 on comparisons with prior works in our paper, and highlight the contributions compared with prior works Acharya et al. (2020) and Yuan et al. (2024).

---

### Official Review · Reviewer_5nRR · 2025-03-24

**Overall Recommendation:** 5

**Summary:**

The authors consider the communication-constrained problem of nonparametric function estimation, in which m distributed terminal observe each n i.i.d. samples drawn from some distribution parameterized by f, and each can send a message of L bits to a central decoder, which is then interested in estimating f under a minimax quadratic risk. Previously sent messages are observable to all terminals (a blackboard model). The work characterizes the minimax optimal rates under certain regularity assumptions for all regimes, and identify phase transitions of
the optimal rates as a function of (m,n).
The upper bound is an algorithm that is constructed of two layers: an outer layer that converts the original nonparametric distributed estimation problem into a distribution estimation problem, and an inner layer that estimates the parametric distribution. This is achieved by a clever reduction and invoking protocols for the parametric density estimation problem. The layer bound is information theoretic and consists on analysis the balls and bins model.

**Claims And Evidence:**

The claims made in the submission are supported by clear and convincing evidence.

**Essential References Not Discussed:**

To my knowledge, there are no essential works not currently discussed in the paper.

**Experimental Designs Or Analyses:**

There are no  experimental designs.

**Methods And Evaluation Criteria:**

The proposed methods make sense for the problem at hand.

**Other Comments Or Suggestions:**

Some theorems in the main body are actually corollaries (e.g., 2.7, 2.12),under Remark 2.3 N_ess = (2^l m n) ^{(2r+1)/(2r+2)} \wedge l m, the last paragraph in section 3 is unclear and should be expanded or moved to section 5, at the bottom of p.8 (last equation), \log ^ m should be \log ^4 n.

**Other Strengths And Weaknesses:**

The results are a significant contribution to the field of distributed estimation with communication constraints. The minimax risk is characterized for a large class of estimation problems and all ranges of (m,n) and the proofs are technically complex and require innovative combination of existing results.
There are some weaknesses in the way the paper is written. I understand there are space limitations, but I still think elaborating on the upper and lower bounds should greatly improve the paper, in particularly the upper bound and section 3 on Wavelets, which is very terse.

**Questions For Authors:**

I don't have any important questions for the authors.

**Relation To Broader Scientific Literature:**

The authors fully solve the problem by characterizing the optimal rates for all regimes of (n,m), whereas previously only specific cases were considered.

**Theoretical Claims:**

I checked the correctness of the proofs of the theoretical claims in the main paper and up to Appendix C.

---

> ### Author Rebuttal · Authors · 2025-03-29
>
> Thanks for your very positive assessment and the suggestions. We are going to revise our paper according to your suggestions in the following aspects.
>
> 1.  Adding more details to Sections 3 and 4 on wavelets and upper bounds within the space limitation. The last paragraph of Section 3 that provides preliminary results for the lower bound will be moved to Section 5.
>
> 2.  Modifying our paper following the other minor suggestions. The only exception is Remark 2.3, where we consider the case $n = 1$, hence $N_{ess}$ is actually $(2^l m) ^{\frac{2r+1}{2r+2}} \wedge m$ instead of $(2^l mn) ^{\frac{2r+1}{2r+2}} \wedge ml$ (please also note the $\wedge mn$ term in the definition (2) of $N_{ess}$).

---

### Decision · Program_Chairs · 2025-05-01

**Decision:**

Accept (poster)

**Comment:**

This paper studies the distributed estimation problem in a nonparametric setting. Although there has been a rich line of literature on this topic, this paper focuses on the challenging scenario where each terminal (i.e. distributed node) has multiple observations, whereas most prior work assumes 1 observation per terminal. For general Sobolev balls, this paper completely characterizes the minimax rate of distributed estimation (up to log factors) and shows exact phase transitions from sparse to dense samples per terminal. Both upper and lower bound analyses are technically sound and significant.

All reviewers unanimously appreciate the significance of this paper. The main concern is on a more detailed comparison with (Zaman and Szabo, 2022), with a more detailed technical discussion on the new Lemma 5.5 and 5.6 in the lower bound proof. I happily recommend acceptance.